# Relationship of insect biomass and richness with land use along a climate gradient

Johannes Uhler[1], Sarah Redlich [2], Jie Zhang [2], Torsten Hothorn[3], Cynthia Tobisch[4], Jörg Ewald[5], Simon Thorn[1], Sebastian Seibold [6,7], Oliver Mitesser[1], Jérôme Morinière[8], Vedran Bozicevic[8], Caryl S. Benjamin[9], Jana Englmeier[1], Ute Fricke [2], Cristina Ganuza[2], Maria Haensel [10], Rebekka Riebl [10], Sandra Rojas-Botero [11], Thomas Rummler[12], Lars Uphus[9], Stefan Schmidt [13], Ingolf Steffan-Dewenter[2] & Jörg Müller [1,14✉]

Recently reported insect declines have raised both political and social concern. Although the declines have been attributed to land use and climate change, supporting evidence suffers from low taxonomic resolution, short time series, a focus on local scales, and the collinearity of the identified drivers. In this study, we conducted a systematic assessment of insect populations in southern Germany, which showed that differences in insect biomass and richness are highly context dependent. We found the largest difference in biomass between semi-natural and urban environments (−42%), whereas differences in total richness (−29%) and the richness of threatened species (−56%) were largest from semi-natural to agricultural environments. These results point to urbanization and agriculture as major drivers of decline. We also found that richness and biomass increase monotonously with increasing temperature, independent of habitat. The contrasting patterns of insect biomass and richness question the use of these indicators as mutual surrogates. Our study provides support for the implementation of more comprehensive measures aimed at habitat restoration in order to halt insect declines.

[1] Field Station Fabrikschleichach, Department of Animal Ecology and Tropical Biology, Julius-Maximilians-University Würzburg, Würzburg, Germany. [2] Department of Animal Ecology and Tropical Biology, Julius-Maximilians-University Würzburg, Würzburg, Germany. [3] Epidemiology, Biostatistics and Prevention Institute, University Zürich, Zürich, Switzerland. [4] Institute of Ecology and Landscape, Weihenstephan-Triesdorf University of Applied Sciences, Freising, Germany. [5] Botany & Vegetation Science, Faculty of Forestry, Weihenstephan-Triesdorf University of Applied Sciences, Freising, Germany. [6] Ecosystem Dynamics and Forest management Group, Technical University of Munich, Freising, Germany. [7] Berchtesgaden National Park, Berchtesgaden, Germany. [8] AIM – Advanced Identification Methods GmbH, Leipzig, Germany. [9] TUM School of Life Sciences, Ecoclimatology, Technical University of Munich, Freising, Germany. [10] Professorship of Ecological Services, Bayreuth Centre of Ecology and Environmental Research (BayCEER), University of Bayreuth, Bayreuth, Germany. [11] Chair of Restoration Ecology, Technical University of Munich, Freising, Germany. [12] Institute of Geography, University of Augsburg, Augsburg, Germany. [13] SNSB-Zoologische Staatssammlung Muenchen, Munich, Germany. [14] Bavarian Forest National Park, Grafenau, Germany. ✉email: joerg.mueller@npv-bw.bayern.de

Insects contribute many functions and services, both in natural and in managed ecosystems. Some of their contributions, such as pollination and pest suppression, are of high economic value[1–4]. The ongoing loss of insect biodiversity has therefore gained global attention[5], with recent reports on declines in insect abundance, richness, and particularly biomass[6–8] provoking rapid actions by policymakers. With agricultural intensification considered as a major driver of insect decline[9], new laws and strategies have been implemented in many countries in order to provide additional resources for farmland species, improve the quality of landscape habitats, and reduce pesticide use in agricultural production[10].

Yet, despite evidence of the negative effects of pesticides[7], habitat loss[11], and low habitat quality[12] on insect communities, whether agricultural intensification is indeed the main culprit in the widespread insect decline is unclear, as losses have occurred not only in agricultural but also in forested environments[6,13]. Moreover, land-use intensity studies have been largely restricted to forests and grasslands[13] or have focused on specific land-use effects, such as those within agricultural or urban areas[14,15]. An additional consideration is the impact of other potential drivers of insect decline, such as climate change, that act simultaneously with land-use intensification[16,17]. However, as is evident from existing time series, it can be difficult to separate the effects of these drivers if they change simultaneously over time[5]. Similarly, the results of landscape-level studies aimed at disentangling the effects of climate and land use are problematic, since land-use gradients generally vary along climate gradients[17]. At the same time, meta-analyses of time series have been hindered by the fact that existing research has primarily focused on only a few taxa[18] or on certain facets of biodiversity, such as biomass or species richness[9,19], although these parameters may respond differently to different drivers of global change[9,20,21] and may be influenced by the spatial scale considered. Space-for-time studies cannot replace long-term time series, but they are complementary, helping to fill gaps in long-term data series in the short term. Moreover, space-for-time studies allow the inclusion of a large number of sampling locations and therefore assessments of the combined impacts of climate and land-use intensity across all land-use types, from semi-natural to agricultural to urban.

Insects respond not only to macroclimate and large-scale land use, but also to local habitat conditions[22] as well as humidity and temperature[23]. For some aspects of biodiversity, the effects thereof may mask or outweigh those of macroclimate and landscape. Changes in insect populations may therefore result from changes in climate, local temperatures, and local or landscape-scale land use, which thus requires that these drivers are studied in concert. While space-for-time is not the best study design, it is generally the best available design, as sufficiently replicated long-term studies do not exist and implementing them now would not provide timely results. Elucidating the roles of habitat loss and climate change requires an approach that covers independent gradients of land use and climate, such that the effects of anthropogenic habitat alterations can be disentangled from those of climate change, local habitat, and temperature while facilitating investigations of the potential interactions of the various drivers[17].

We asked whether the magnitude of the progressive decline in insect populations[6,13] could also be discerned spatially, between habitat types, landscape types, or combinations thereof. Thus, in a space-for-time substitution approach we set up 179 Malaise traps in 2019 along a local land-use gradient of increasing intensity, ranging from forests, to meadows, to arable fields, and finally to settlements, thereby including the full range of land-use intensities in temperate Europe[24] (Fig. 1). The level of land-use intensity in the surrounding landscape (ranging from semi-natural to agricultural to urban) was also considered[24]. Based on prior knowledge of insect ecology and our study design, we refer to statistical relationships between the environment (e.g. land use, climate) and insects as evidence of effects of the former and responses of the latter. The analyzed sites are spatially distributed over 400 km and cover an elevational gradient of ~1000 m, creating a largely land-use-independent macroclimatic gradient with a mean annual temperature range between 5 and 10.3 °C and an annual precipitation of 550–1961 mm.

Local temperature and humidity were recorded at each trap for each sampling period. The traps were emptied twice a month throughout the vegetation period (eight sampling campaigns in total). The biomass of the captured arthropods (referred to in the following as "insects") was measured, and metabarcoding was used to quantify the arthropod taxa[25] and the richness of the BINs (barcode index number, referred to in the following as "total richness"), which strongly correlates with biological species[26]. In addition, the richness of metabarcoding-identified species especially relevant for conservation (red-listed species) was assessed using a compilation of Red Lists for insects across all lineages.

Generalized additive models were fitted to simultaneously test for the effects of climate variables (annual mean temperature and precipitation, local temperature and humidity) during sampling and the land-use categories (local and landscape scales) on biomass and species richness. In all models, the pure effect of the season was taken into account by adding it as a smoothed fixed effect; the length of the individual sampling period was included as an offset to control for variable sampling durations. To account for the spatial arrangement within and between grids and for repeated measurements on each plot, a correlated plot-specific intercept was applied. Interactions between local temperature and land-use variables were also tested as were the non-linear effects of climate and the local temperature and humidity. Finally, the partial effect coefficients of the local and landscape-level land-use categories and their combinations were extracted from the models to determine whether the temporal decline reported in previous studies can be explained by changes in biomass and species richness comparable to those determined along the land use and climate gradients defined in this study.

Overall, we found that both insect richness and biomass increase monotonously with increasing temperature, independent of habitat. Regarding the effects of land use, we found the largest difference in biomass between semi-natural and urban environments (−42%), whereas differences in total richness (−29%) and the richness of threatened species (−56%) were largest from semi-natural to agricultural environments. The contrasting responses of biomass variation and BIN richness to local and landscape-scale land use point to differential effects of shifts in land use on insect populations, with ongoing urbanization leading to a decline in biomass, and conversion to agriculture to a decline in species richness.

## Results

Based on 1293 insect samples, the average biomass was ~2.6 g/day, with a strong hump-shaped partial effect of season with a clear peak in the second half of June (Fig. 2a). From the metabarcoding results of 510 samples, 7589 BINs were identified (Supplementary Fig. 1). These could be assigned to ~5900 taxonomically described species, ~700 of which belonged to red-listed species. Both the total richness and the richness of red-listed species showed a convex response to the partial effect of season, with the highest values occurring at the beginning of the sampling period (May) and the lowest values around early July (Fig. 2). All three measures of insects increased linearly with increasing local temperature, but only total richness increased with increasing

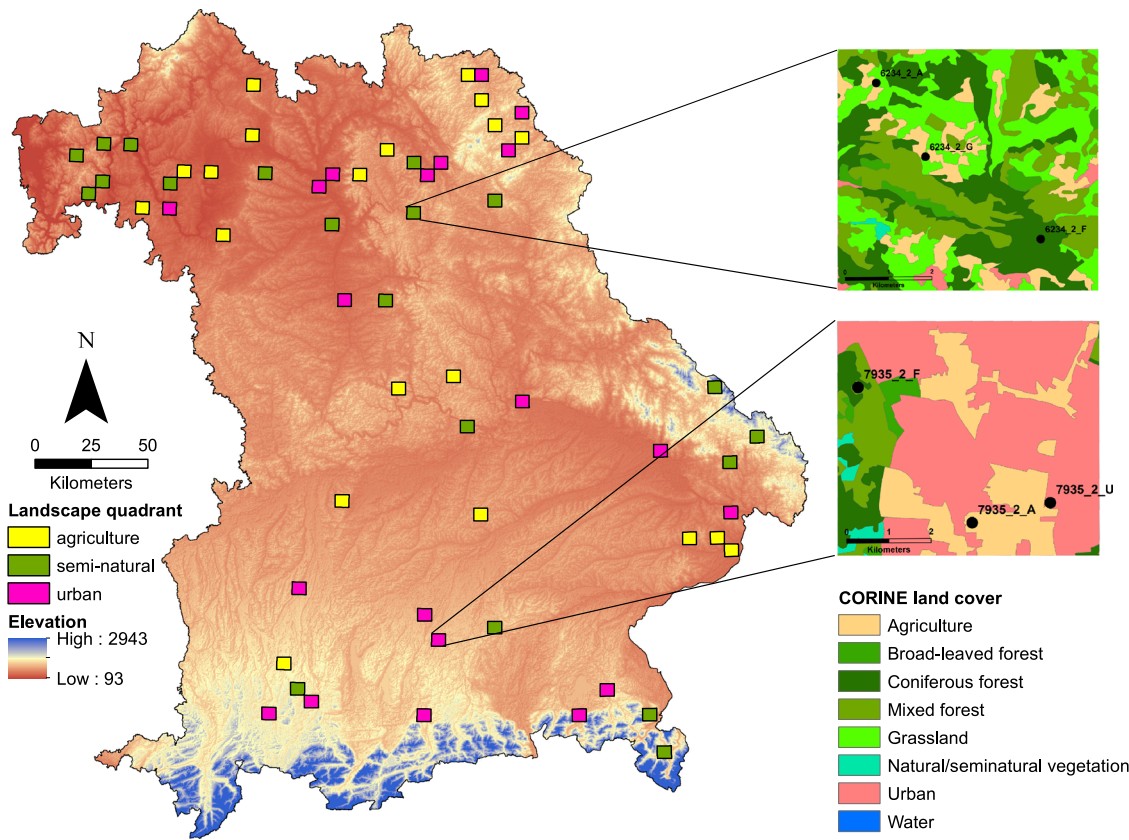

**Fig. 1 Overview of the study design and the distribution of the 60 landscapes across the study region.** The 60 quadrants were selected over gradients of elevation and land-use type within the federal state of Bavaria. The colors show the elevation. The insets show examples of a semi-natural and an urban-dominated landscape. The black dots represent the three plots set up in different habitat types.

humidity (Fig. 2b, c). It is important to note that we separated the change over the growing season into the pure partial effects of season and local temperature and humidity. When local temperature and humidity were excluded from the models, both biomass and species richness followed a hump-shaped curve, even though the shape was a lot less pronounced for species richness compared to biomass (Supplementary Fig. 2).

Macroclimate variables had significant positive effects on total richness and the richness of red-listed species but not on biomass (Table 1). Tests for non-linear response to macro- and microclimate variables, such as the negative effects of high temperatures, did not yield humped-shaped or concave curves.

At the scale of local land use, biomass was highest for forests and significantly lower for arable fields followed by settlements (Table 1, Fig. 3a). Total richness and the richness of red-listed species were also highest for forests and significantly lower for settlements and meadows followed by arable fields (Fig. 3a). At the landscape scale, biomass was highest in agricultural landscapes and significantly lower in urban landscapes, whereas red-listed richness was highest in semi-natural and lowest in agricultural landscapes (Table 1, Fig. 3b). Quantifying the combined effects of local and landscape-level land use showed that the largest difference in biomass (−42%) occurred between forests in agricultural landscapes and settlements in urban landscapes (Fig. 3c). The largest difference in the total richness of all taxa (−29%) was between forests in urban landscapes/settlements in semi-natural landscapes and arable fields in agricultural landscapes (Fig. 3c). For red-listed species richness, the largest difference (−56%) was between forests in semi-natural landscapes and arable fields in agricultural landscapes (Fig. 3c). Modeling the total biomass per sample and taxon over all habitats showed that

specific estimates of total biomass increased consistently only for the BIN richness of three taxa (Lepidoptera, Orthoptera, and Diptera) across all habitats (Fig. 4).

Tests for relationships between local temperature and land use failed to reveal a significant interaction, indicating independent effects of these two parameters on insect biomass and richness within our study area. Finally, land use and local climate were further explored by modeling the local temperature at each trap, using the same approach as for the insect measures. Strong effects of land use were determined, even after controlling for the season, elevation, and the geographic location of the traps, with increasing temperatures from forests to meadows to arable fields and to settlements, and similarly from semi-natural and agricultural landscapes to urban landscapes (Supplementary Table 2).

## Discussion

Our approach provides data on species richness across independent gradients of land-use intensity and climate. Furthermore, by combining Malaise traps and DNA-metabarcoding, our work is not limited to single factors such as biomass measurements or assessment of single taxa to reveal drivers of insect communities. We found the lowest species richness in arable fields embedded in agricultural landscapes, and the lowest biomass in settlements embedded in urban landscapes. The effects of land-use type were independent of those of local temperatures and climate. Biomass and richness measures differed according to land-use intensity. Our study recorded a difference in insect biomass of 42% from semi-natural to urban environments, but no difference from semi-natural to agricultural environments. This appears to be in contrast with the results documented in a similar analysis[6], which showed a temporal decline in insect biomass of >75% in small,

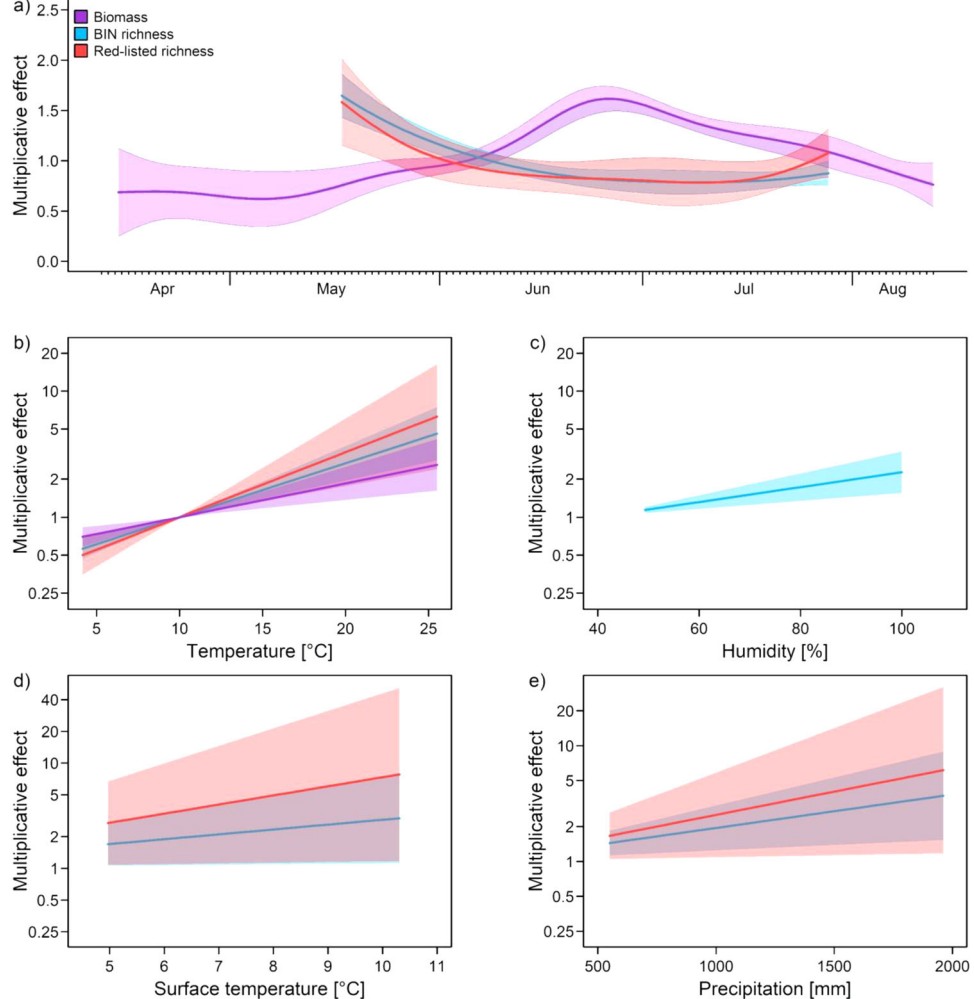

**Fig. 2 Partial effects of season and climate parameters on biomass, total BIN richness and richness of red-listed species.** Partial effects of (**a**) season, (**b**) local temperature, (**c**) humidity (both data logger), (**d**) long-term mean annual near-surface temperature (1991–2020), and (**e**) long-term mean annual precipitation (1991–2020) on biomass, total richness of barcode index numbers (BINs), and the richness of red-listed species collected with Malaise traps in 179 plots. Partial effects from generalized additive mixed models (for details, see Table 1) were controlled for elevation, the geographic location of the traps, and land use (for the latter, see Fig. 3). Note that richness was determined for only three of the eight sampling campaigns. (a) partial effect of season, as a smooth term acting multiplicatively on the expected outcome per time unit. Multiplicative effects for local temperature and humidity were estimated using baselines of 10 °C and 40%, respectively. Error envelopes depict standard errors below and above the estimated mean responses.

protected areas surrounded by an agricultural landscape. Interestingly, in Hallmann et al.[6], the few plots in semi-natural landscapes also showed a similar temporal decline as those in agricultural landscapes (Supplementary Fig. 3b). On the other hand, the variation in total BIN richness matched the magnitude of the temporal decline (~35%) determined over a decade in grasslands and forests by Seibold et al.[13]

The hump-shaped seasonal pattern of biomass and associated daily biomass values were in accordance with the time series of Hallmann et al.[6], thus demonstrating the comparability of our space-for-time approach with approaches based on time series (Supplementary Fig. 3). However, the contrasting phenological patterns of biomass and total BIN richness after controlling for temperature are evidence that both facets of biodiversity might respond differently, with biomass more strongly driven by pure season, e.g. via plant phenology or day-length, and BIN richness more dependent on local temperature. Divergent responses of biomass variation and species richness have already been described in temporal studies of insects in freshwater systems[27] and nocturnal moths in the United Kingdom[19,28,29], but not in studies of terrestrial arthropods, including those recorded in

comprehensive datasets of hyper-diverse orders such as Diptera and the Hymenoptera.

The positive relationships between local temperature and biomass variation and BIN richness were consistent with earlier results[6,20] and can be explained (1) by the higher activity of species at higher temperatures, which increases the likelihood of trapping[30] and (2) by the fact that insects are ectothermic organisms, i.e., their metabolism is enhanced by increasing temperatures, which in turn can lead to higher reproduction and survival rates and thus to larger populations[31]. Our additional analyses on the negative effects occurring at the highest temperatures did not provide any such indications for our three measures. Moreover, insects, and in particular many endangered insect species in Central Europe, are thermophilic[32], which would explain the observed response of total BIN richness, and especially the very steep response of the richness of red-listed species, to local temperature.

Despite the positive or neutral effect of macroclimate and the consistently positive effect of local temperature on insect biomass and BIN richness, global warming can cause shifts in insect communities that threaten biodiversity in specific biomes or

**Table 1 Effects on insect biomass and diversity.**

| Predictors | Biomass | | | | | BIN Richness | | | | | RL Richness | | | | |
|---|---|---|---|---|---|---|---|---|---|---|---|---|---|---|---|
| | Estimates * 10³ | Std. error * 10³ | t-value | p | Coefficient | Estimate (Log-mean) * 10³ | Std. error * 10³ | z-value | p | Coefficient | Estimate (Log-mean) * 10³ | Std. error * 10³ | z-value | p | Coefficient |
| (Intercept) | 435.6 | 917.3 | 0.475 | 0.635 | 1.545 | −922.7 | 730.66 | −1.263 | 0.206 | 0.397 | −4480.9 | 1408.6 | −3.181 | **0.0014** | 0.011 |
| *Local land use: Forest* | | | | | | | | | | | | | | | |
| Meadow | −47.23 | 33.42 | −1.414 | 0.157 | 0.953 | −208.0 | 29.10 | −7.147 | **<0.001** | 0.812 | −266.21 | 56.49 | −4.712 | **<0.001** | 0.766 |
| Arable field | −111.1 | 34.58 | −3.213 | **0.0013** | 0.894 | −313.3 | 30.30 | −10.339 | **<0.001** | 0.731 | −616.00 | 61.39 | −10.03 | **<0.001** | 0.540 |
| Settlement | −250.5 | 41.61 | −6.019 | **<0.001** | 0.778 | −113.02 | 34.11 | −3.313 | **<0.001** | 0.893 | −203.34 | 64.92 | −3.132 | **0.0017** | 0.815 |
| *Landscape land use: Semi–natural* | | | | | | | | | | | | | | | |
| Agricultural | 80.32 | 40.10 | 2.003 | **0.045** | 1.083 | −39.14 | 34.79 | −1.125 | 0.260 | 0.961 | −155.9 | 67.13 | −2.323 | **0.020** | 0.855 |
| Urban | −94.80 | 44.39 | −2.136 | **0.032** | 0.909 | −59.77 | 34.88 | −1.714 | 0.086 | 0.941 | −79.68 | 67.53 | −1.180 | 0.238 | 0.923 |
| Long-term mean annual precipitation | −0.021 | 0.330 | −0.066 | 0.947 | - | 0.666 | 0.222 | 2.912 | **0.004** | - | 0.925 | 0.431 | 2.146 | **0.031** | - |
| Long-term mean annual near-surface temperature | −10.74 | 66.68 | −0.161 | 0.872 | - | 106.1 | 4.860 | 2.185 | **0.028** | - | 199.24 | 93.84 | 2.123 | **0.033** | - |
| Local temperature (4.1–25.5 °C) | 61.30 | 15.50 | 3.955 | **<0.001** | - | 98.18 | 16.05 | 6.116 | **<0.001** | - | 118.37 | 31.58 | 3.748 | **<0.001** | - |
| Local humidity (25.5–100%) | −4.039 | 3.380 | −1.195 | 0.232 | - | 13.62 | 3.235 | 4.211 | **<0.001** | - | −7.258 | 6.447 | −1.126 | 0.260 | - |
| Observations | 1293 | | | | | 510 | | | | | 510 | | | | |
| R²/sp.R² | 0.552 | | | | | 0.268 | | | | | 0.483 | | | | |

The results of generalized additive mixed models for insect biomass (8 sampling campaigns), richness of barcode index numbers (BINs), and red-listed species (3 sampling campaigns) using land-use categories on local and landscape scales as well as long-term mean annual precipitation (1991–2020) and long-term mean annual near-surface temperature (1991–2020), local temperature, and humidity as fixed linear effects, day as a smoothed effect, and space as a random effect. To control for differences in the sampling periods, log(sampling days) was used as an offset. Multiplicative-partial effect coefficients of land use were extracted using family = gaussian(link = "log") for biomass, and family = negative binomial for richness. Season and the geographic location of the traps yielded the following p-values for biomass ($p < 0.001$, $p < 0.001$), BIN richness ($p < 0.001$, $p < 0.001$), and red-listed richness ($p = 0.012$, $p < 0.001$).

elevations[7,8,33], by a mismatch between host plant and insect phenology[34,35] or by the trait-specific responses of species to climate variations, as shown for butterflies in California[33]. Nevertheless, the responses of insect populations and insect diversity to climate change are poorly understood, such that clear patterns, with distinct winners and losers, can still not be discerned[33]. In addition, insect responses to climate change are geographically variable and likely to be disproportionally higher at higher latitudes and elevations or in hot tropical or Mediterranean areas[33]. However, it is precisely the large topographic variation of mountains that may offer climate pockets that act as refugia, thus allowing insects to survive during periods of extreme climatic conditions or climate variation[33,36]. Our study supports this possibility, by showing that the responses of total insect richness, the richness of red-listed species, and biomass to higher local temperatures in a cultivated landscape in Central Europe (mean annual temperature of ~5 to 10 °C and annual precipitation between 550 and 2000 mm) are consistently positive. A further rise in temperature, as expected in the near future, poses a high risk of pushing more insect species in our study area to their thermal limits and even to extinction[37].

The clear biomass patterns which we show indicate a continuous change of biomass from forests to arable fields and further to settlements, of total BIN richness from forests to arable fields, and of red-listed species richness from forests to meadows and arable fields. This underlines the importance of forests as a backbone of insect diversity in cultivated landscapes, and particularly of forest gaps, which are rich in species within forests[13,38]. Our study is the first to our knowledge to directly compare forests (and forest gaps) with agricultural and urban habitats. Comparable studies using standardized insect sampling across a broad range of land-use types are rare, but data on the biomass of moths obtained by light trapping in different habitats over many decades[19] are consistent with our findings and indicate a general pattern that is independent of the sampling method. At the landscape scale, we found biomass was highest in agricultural landscapes and lowest in urban landscapes, whereas red-listed richness was highest in semi-natural landscapes, followed by urban landscapes and lowest in agricultural landscapes. Although we could not confirm the negative effects of agricultural landscapes on biomass, as described by Hallmann et al.[6], our results are in line with those of Seibold et al.[13], who reported negative effects of surrounding arable fields on the temporal trends in grasslands in terms of species richness but not insect biomass.

The contrasting pure seasonal patterns of biomass variation and BIN richness, as well as their different responses to land use, may have methodological or biological causes. A possible methodological reason for the low partial effects of season on BIN richness during summer but high partial effects on total biomass is that high insect biomass occurs particularly during periods of high temperatures, which would have increased evaporation of the ethanol used for preservation, accelerating the degradation of DNA. Similar effects were shown for samples stored over long periods[39] of time. However, in our study, the collection bottles contained sufficient amounts of ethanol such that a methodological effect due to ethanol evaporation was unlikely. Moreover, high temperatures and not the pure seasonal effect better explained the higher BIN richness in this study. A second methodological reason for the lower BIN richness is that small species are often "overlooked" in biomass-rich samples[40–42]. To avoid this problem, we divided each sample into two fractions (small and large species) and sequenced them separately. With the exclusion of these methodological reasons, the most likely explanation for our findings is a biological one related to the composition of the samples. An increase in large species in certain habitats or at a certain time of year could influence biomass but not necessarily the total number of species. However, our additional models of total biomass using the

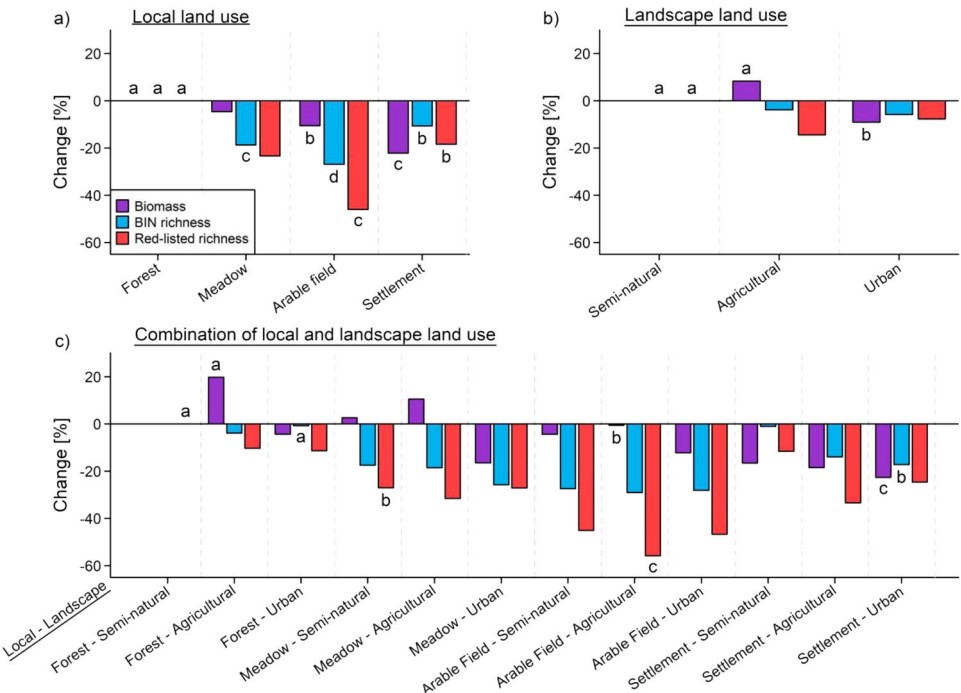

**Fig. 3 Partial effects of land use on insect biomass and diversity (total BIN richness and the richness of red-listed species).** The results of generalized additive mixed models for insect biomass (8 sampling campaigns, $n = 1293$), richness of barcode index numbers (BINs), and red-listed species (3 sampling campaigns, $n = 510$) using land-use categories on local and landscape scales. The displayed values are based on comparisons with (**a**) the local land-use type forest, (**b**) semi-natural landscapes, and (**c**) the combined local land-use type forest in semi-natural landscapes. For model parameters, see Table 1. For additional information see annotated code below. Significance was tested by multiple post-hoc comparisons using *glht* (R package multcomp[64]). Different letters indicate significant differences ($p < 0.05$) between categories. Note that only the first significant entry for a predictor is shown, subsequent entries between land-use categories, even if significant, were omitted for clarity.

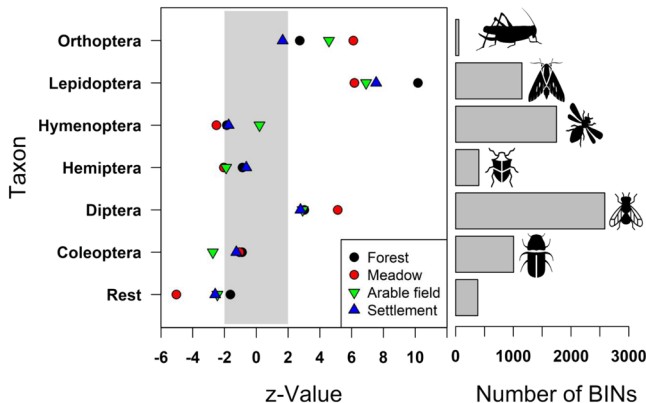

**Fig. 4 Taxonomic group and habitat-specific standardized estimates of BIN richness to explain total biomass in a generalized linear mixed model.** The horizontal bars on the right show the total number of BINs per group. Values for BIN richness and total biomass were log-transformed. The vertical gray bar on the left indicates the area outside of which effects are significant.

BIN richness of the most important taxonomic orders as predictors provided an important clue. Across all habitats, biomass variation was best explained by the increase in BIN richness of three species groups, Orthoptera, Lepidoptera, and Diptera. Of the diverse taxa Coleoptera, Hymenoptera, and Diptera, only the BIN richness of the latter positively affected total biomass, and it was principally the richness of the two groups with many large species (Orthoptera and Lepidoptera) driving the pure seasonal effect. This can be explained by the fact that Lepidoptera abundance peaks in July[43], thus coinciding with the higher abundances of most species of

hemimetabolous Orthoptera during the summer[44], and therefore accounting for the purely seasonal peak of insect biomass in summer.

The contrasting responses of biomass variation and BIN richness point to differences in the respective mechanisms. Insect biomass is positively related to productivity and is thus highest in agricultural landscapes and in forests habitats embedded in agricultural landscapes managed to maximize plant productivity and continuous plant biomass[45,46]. Insect biomass is lowest in urban environments, where productivity is limited due to a high percentage of sealed areas without vegetation. However, insect biomass along gradients of urbanization has been poorly investigated[47] such that large differences in the negative effects of urbanization on the abundances of different taxonomic groups cannot be ruled out[48]. Moreover, urban areas include additional potential stressors, such as light pollution, that might also negatively affect insect biomass[49]. In contrast to biomass, the richness of all taxa and of threatened species was relatively high in urban habitats. This was especially the case for urban habitats embedded in semi-natural landscapes, although a similar species richness may occur through the interplay of semi-natural habitats with green spaces characterized by a highly variable design and management[50] as well as with the natural but also anthropogenically enhanced plant diversity of urban areas[47,51,52].

The lowest BIN richness generally observed in our study, in arable fields embedded in agricultural landscapes, is consistent with the results of a recent meta-analysis of insect time series[9]. In that study, the temporal declines in insect populations of terrestrial invertebrates were largest in regions with generally high agricultural land-use intensity, such as Central Europe and the American Midwest. Our direct comparison of different land-use types independent of gradients of macro- and microclimate suggests that the strong declines in insect richness reported for several taxa[5] are indeed

driven by intensive agriculture and the associated homogenization of the landscape[53], not by urban environments. To assess the significance of our two main results on biomass and species richness, however, it is necessary to consider the proportions of the land-use types in question. In our study, agricultural land comprised 48% of the area whereas settlements accounted for ~12%. Since habitat amount is a fundamental parameter for insect populations, it must also be taken into account in a country-wide strategy[11].

Our finding of a lack of significant interactions between the highly significant local temperature and land use contrasts in part with the previously reported strong effects resulting from the interaction between land use and climate along the elevational gradient of the Kilimanjaro. That finding implied that land-use effects are mediated by climate, especially at high elevations[17]. Interaction effects between land use and climate may thus occur mainly within more extreme climates[54] rather than within the temperate climate exemplified by our study region. By considering macroclimate and the directly measured local temperature and humidity as well as land use, we were able to show that pure land-use effects, when evaluated as habitat effects controlled for local temperature and humidity, strongly influence insect populations. However, despite the increasing awareness among scientists and urban planners that land use at local and landscape scales impacts not only insects but also local climate, the implications have mostly been ignored in international climate negotiations[55]. Trees, with the reduced local temperatures offered by their canopy layer[56] and their hosting of a high species richness of insects, as shown in our study, are thus of particular importance as refuges for insect diversity in temperate zones.

By covering the full range of land-use intensities along the climate gradient of a typical cultivated region and measuring both insect biomass and total insect richness, our study's methodology provided mechanistic insights into the changes of insect populations in areas where a meta-analysis identified the most severe population declines[9]. Nevertheless, additional studies should focus on biomes other than the cultivated landscapes of the temperate zone, such as cold boreal, dry Mediterranean, or hot tropical areas. Here, the different characteristics of the biome may result in land-use intensification being of less importance than climate change. In addition, the use of metabarcoding to identify all insects within a sample broadens the range for similar space-for-time studies. In contrast to well replicated, standardized time-series data that may require decades to generate the information needed to guide conservation actions, space-for-time approaches covering full gradients of land use and climate are a viable option to identify the drivers of insect decline and thus provide timely information for decision-makers; however, replications from several years should be included to take into account the effects of extreme events.

The weak effect of climate variables on insect biomass but the consistently positive effect of local temperature on biomass variation and BIN richness suggests that, at least within the climate range of our temperate study region, the recent warming that has led to higher local temperatures should promote insect biomass and species richness. However, further warming, extreme heat, and drought events may negatively affect biodiversity, although non-linear responses can be expected in other climates or across longer gradients. Moreover, the strong dependency of local temperature on land use indicates that changes in land use impact local climate conditions, such as by accelerating temperature increases in agricultural and urban regions. The contrasting responses of biomass variation and BIN richness to local and landscape-scale land use point to differential effects of shifts in land use on insect populations, with ongoing urbanization leading to a decline in biomass, and conversion to agriculture to a decline in species richness. Based on our results, we recommend that actions aimed at preventing further insect decline should focus on (1) increasing insect biomass, for example by improving "green" habitats in urban environments[57] and reducing the extent of vegetation-free sealed surfaces and (2) stopping the ongoing loss of species, by adapting agri-environmental schemes and promoting habitats dominated by trees, even in urban environments.

## Materials and methods

**Study design and land use data.** This study was conducted in 2019 as part of the LandKlif project (https://www.landklif.biozentrum.uni-wuerzburg.de), in Bavaria, Germany. A detailed description can be found in Redlich et al.[24]. Based on a grid cell system (TK 25[58]) covering all of Bavaria, we selected 60 landscapes of ~5.8 km × 5.8 km, creating gradients of land use and climate[24]. First, five climate zones were defined based on the mean annual temperature over 30 years (1981–2010): <7.5, 7.5–8, 8–8.5, 8.5–9, and >9 °C[24]. Land use at the level of a quadrant, referred to as the landscape level, was classified into three categories based on Corine land cover data from 2012[24]: semi-natural, defined as natural vegetation (forest, grassland, and natural-semi-natural) >85% and forest >50%; agricultural, defined as arable fields >40%; and urban, where the urban area comprised >14%, which is more than the region-wide average. For each of the 15 combinations of climate zone and land-use types, four quadrants were selected, for a total of 60 quadrants (Fig. 1). The final 60 quadrants were those with the least amount of spatial autocorrelation and logistic restraints (e.g., distance to the nearest motorway). Within each quadrant, three survey plots were selected based on the three most common land-use types of the quadrant (of the four types considered in this study: forest, meadow, arable field, settlement), the least amount of correlation between landscape configuration and composition, and the least amount of spatial overlap (2 km apart if possible).

Suitable survey plots were identified using remote sensing information. In general, sampling conditions were standardized by selecting a grassy area of (if possible) 0.5 ha located within the local land-use type (i.e., within forest or settlement, between fields, on grassland) or close enough to the local land-use type to capture its effects on insect biomass and species richness. After permission was granted by the local landowner, a core 3 × 30 m strip at the center of the 0.5-ha grassy area was chosen for the experiment. This core strip was left unmanaged for the duration of the experimental period, as its management would have interfered with the traps.

Survey plots in forests were established in forest openings where sunlight was able to reach the ground for most of the day (diameter of forest clearing at least the height of local trees), to minimize bias in activity trapping[6], and at least 50 m from the forest edge. Where possible, broadleaf rather than coniferous forest was chosen because the former is the natural forest type in the study region. Survey plots in settlements were set up on available green areas, e.g., vacant lots or public parks, and >50 m away from main roads, if possible. Meadow plots were established on grasslands, as far away as possible from arable fields (at least 50 m). Arable field plots were established on green strips or barren land next to conventionally managed fields, to avoid collision with farming machines. Plots next to oilseed rape were preferentially selected to permit pollinator exclusion experiments. In total, 179 plots were established.

**Arthropod sampling.** At each plot, one Malaise trap was installed in the plot center. To avoid restricted access to the traps by small woody features (e.g. shrubs), the traps were oriented orthogonal to the edge of patches with high vegetation. In plots without high vegetation, the traps were randomly positioned. In addition, vegetation in close proximity to the trap entrance was kept low throughout the season. The Malaise traps were based on the Townes Malaise trap model, albeit with a black roof and a slightly smaller size (dimensions of the capture area: height front: 0.90 m; height rear: 0.60 m; length: 1.60 m; conventional Malaise trap dimensions: height front: 1.10 m; hight rear 0.90 m, length: 1.75 m). Ethanol (80%) was used as the capture fluid to ensure the preservation of the DNA for barcoding. The traps were activated mid-April and emptied every 2 weeks until mid-August, for a total of eight complete sampling campaigns on all plots. Due to logistical constraints, the individual sampling period was variable. Of 1432 possible samples, 93 could not be obtained for reasons related to the late start of sampling, collapsed traps, vandalism, or destruction of samples during transport. Three additional samples were destroyed during handling, resulting in biomass data but no further BIN data. Missing climate data (failed dataloggers) resulted in the exclusion of 46 additional samples from the statistical analysis. Thus, a total of 139 samples were excluded from the analysis of insect biomass and 27 from the analysis of BINs (see Supplementary Table 3). Due to financial restrictions, only those samples from the major periods of insect activity, i.e., from the second half of May, the second half of June, and the second half of July, were analyzed by metabarcoding. To measure insect biomass, the samples were collected on a sieve. When the time between two drops of ethanol reached 10 s, the weight of the sample (biomass) was measured using a precision scale. The samples were then sieved through an 8-mm sieve, thereby separating larger and smaller insects. This was done to improve barcoding results and to overcome potential bias regarding the biomass of the small vs. the large fraction, since large-insect size fractions tend to contain fewer species but their biomass and amount of DNA are higher, whereas in small-insect size fractions the species richness is higher but there is less biomass and DNA. Species were identified using CO1-5P (mitochondrial cytochrome oxidase 1; for primers see Supplementary Table 1) DNA metabarcoding (see Supplementary methods) following the laboratory and bioinformatic pipelines reported in Hausmann et al.[25].

To cover the same level of identification for all taxonomic groups, BINs instead of OTUs were used to measure richness, since the latter tends to overestimate richness in some orders, e.g. Orthoptera. The utility of BINs in characterizing formal genetic units independently of an existing classification has been demonstrated[59]. The BIN system of the online platform BOLD is based on a chain of algorithms clustering similar barcode sequences and checking their taxonomic integrity. The BIN clusters match the actual taxonomically identified species at different levels (90–99% COI genetic similarity), depending on the taxa, thus allowing comparisons with studies based on morphological determination. For red-listed species, the number of species reported in one of the categories critically endangered (CR), endangered (EN), vulnerable (VU), or near threatened (NT) as reported in Red Lists for the federal state of Bavaria and for Germany was counted based on a metabarcoding species identification with >97% probability. The consideration of both national and federal scales was necessary to represent the broad taxonomic spectrum of our samples, which is not consistently covered by either of the two Red Lists alone, and was justified by the fact that Bavaria covers ~20% of Germany and includes most landscape types found in the country. Arthropods were collected with the permission and ethical approval of the nature conservation authorities of the governments Upper Franconia, Lower Franconia, Middle Franconia, Lower Bavaria, Upper Bavaria, Swabia, and Upper Palatinate and we complied with all relevant ethical regulations for animal handling and research.

**Climate, local temperature, and humidity**. Local temperature and humidity were recorded with ibutton thermologgers (type DS1923). At each site, one data logger was mounted on a wooden pole at 1.10 m height, facing north. A roof panel provided protection against direct sun exposure. Air temperature and relative humidity were recorded every hour and the hourly measurements then averaged across the sampling periods.

Annual mean temperature and precipitation were calculated for each plot based on gridded monthly datasets with a horizontal resolution of 1 km, using a nearest source to destination approach. Subsequently, long-term averages, i.e., climate normals, and deviations thereof were calculated for the aforementioned quantities and the period 1991–2020. The raw input datasets are provided free of charge by the German Meteorological Service (DWD) and are described in Kaspar et al.[60]. The correlation between climate variables was low for most variables (Pearson's $r < 0.23$), and only moderate for the long-term mean annual average near-surface temperature and long-term mean annual precipitation (Pearson's $r = 0.51$).

**Statistical analysis**. Overall, the data of 1293 samples were used for the analysis of insect biomass and 510 samples for the analysis of BIN richness. Analyses were performed in R v. 3.6.2[61]. Generalized additive models were fitted using the package mgcv[62] to test for the effects of land use and climate on the biomass and species richness of all taxa and of threatened taxa for each sample (Table 1). In all models, the mean day of a trap-specific sampling period was modeled by a smoothed non-linear spline of time, to account for seasonality, an offset of sampling length to control for variation in individual sampling periods, and a correlated plot-specific intercept (geographical position of the plot) to account for the spatial arrangement within and between grids and for repeated measurements per plot. As a predictor of macro-climate, mean annual temperature and precipitation served as a proxy. The mean local temperature and mean humidity at each plot were measured within the sampling windows specific for each trap sample. The latter four variables were ultimately modeled linearly, because the application of smoothed effects did not support non-linear effects as negative effects at high values. The decision was based on the graphical interpretation of smoothed splines as discussed in Heidrich et al.[63].

Predictors for land use included local and landscape land-use types, i.e., referring to the 5.8 km × 5.8 km grid cells. The same models were run, by using a combination of local and landscape land-use types, to test for the dependence of biomass and species richness on the specific landscape context. For biomass, a Gaussian distribution with a log-link was used, and for richness a negative binomial distribution that allowed extraction of the relative values compared to the benchmark land-use category assumed to be the most natural one (Fig. 3). Significant differences between land-use categories were assessed by multiple post-hoc comparisons using the *glht* (package multcomp[64]). The effects of different land-use categories on local temperature, controlled for season, space, and repeated measurement, were modeled as described above with a Gaussian model. The relationship of richness and biomass was investigated by modeling the log-biomass per sample by the log(+1)-transformed richness of the major orders Diptera, Hymenoptera, Coleoptera, and Orthoptera as well as a rest group, using a linear mixed model with a random effect on plot using the functions *cftest* (package multcomp[64]) and *lmer* (package lme4[65]).

**Reporting summary**. Further information on research design is available in the Nature Research Reporting Summary linked to this article.

## Data availability:
The data that support the findings of this study, as well as data from Hallmann et al.[6] used in this study are publicly available on Dryad under https://doi.org/10.5061/dryad.zkh1893bb.

## Code availability
The R script for summary statistics and to generate the graphs is publicly available on Dryad under https://doi.org/10.5061/dryad.zkh1893bb.

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

## Acknowledgements

The authors thank all the landowners across Bavaria for kindly allowing us to work on their land. For technical assistance in the field and laboratory we thank Beate Krischke, Susanne Schiele, Gudrun Krimmer, Jan Weber, Almuth Puschmann, Dragan Petrovic, Linus Krämer. We acknowledge support by the Bavarian Ministry of Science and the Arts via the Bavarian Climate Research Network (bayklif), in particular Ulrike Kaltenhauser, and thank the Bavarian State Forestry, in particular Kay Müller, for their help and cooperation. In addition, we gratefully acknowledge the ability to use datasets from the Deutscher Wetterdienst (DWD). The LandKlif project was funded by the Bavarian Ministry of Science and the Arts.

## Author contributions

J.M., J.U., S.R. S.S. perceived the idea of this manuscript. J.M., I.S.-D., and J.Z. designed the experiment. J.U., S.R., C.T., C.B., J.E., U.F., C.G.V., M.H., R.R., S.R.B., and L.U. collected data. T.H., O.M., J.M., and J.U. analyzed the data. S.T. and J.Z. designed the graphs. J.U., J.M., S.S., S.R., and I.S.-D. wrote the first manuscript draft and finalized the manuscript. All authors commented on the manuscript.

## Funding

## Competing interests

The authors declare no competing interests.
