## [Peer Review File · Nature Communications]

Reviewer #1 (Remarks to the Author):

The authors present a novel study on variation in insects across a number of land-use categories, and investigate phenological and landscape effects on biomass and species richness, the latter as inferred from DNA metabarcoding. They compare land use and climatic effects at two different spatial and temporal scales. They conclude that urbanization and agriculture are major drivers of decline, increased amount of insects along temperature gradients, and that richness and biomass are not exchangeable metrics. I have a number of concerns and suggestions for improvements.

Concerns:

Investigating /sorting out samples out 179x8 traps is a daunting task at best. Use of DNA metabarcoding as surrogate of manual species identification is state of the art among approaches to tackle this logistical constraint. However, using BINs as a surrogate of species richness does not come without a cost, and I have the feeling that the authors have not appreciated (or warned the reader) enough about risks associated with using BINs as surrogate for richness.

I am particularly intrigued by the lack of a seasonal BIN-richness pattern (i.e. not peaking along with biomass). This is at odds with what is currently seen by comparison of biomass with manual species richness determination under the same trapping conditions (see Hallmann et al PNAS 2021). I suspect something is off here. One possibility for example is that DNA has been degraded in the warmer sampling weeks of June-July, affecting the number of BIN-reads. The 80% solution used in this study during trapping is not likely to stay 80% but drop significantly below at hotter days, which undoubtedly affects DNA preservation (see also Marquina et al 2021, PeerJ 9:e10799). At best, this should be discussed in the paper, but I believe it is very worthwhile to investigate this. Eg by measuring ethanol concentration at sample collection date.

I'm surprised to not see a correlation plot between biomass and number of BINs per sample. Even more interesting would be a temporal(seasonal) correlation plot, which would help understanding the data better in relation to the previous comment.

I am confused about what "red-listed species" mean in this context? Species that made it on the red list? Or that they are under some threat-category? Making it on the list does not translate to unfavorable conservation status, as many species have non-threatened status. So why is the sum of red list species -which is what I assume is what has been used- an interesting response variable as opposed to the proportions of the categories? Also, there seems to be a mixing between two different red lists, which resemble different spatial scales (state vs country). This is inappropriate. I notice the use of gradients throughout the paper. However I am under the impression from the materials section that you used land use categories. Not gradients.

Trapping periods seems to be variable (although not specified how much), roughly two weeks. While averaging quantities over the trapping periods is probably ok for biomass analysis, this is certainly not the case for species richness, and by extension the BINs, because richness does not act additively with increasing survey effort (i.e. variable trapping durations per sample). As such, I suggest to add survey time per sample as a fixed, possibly smooth effect, and not as an offset in order to have a better description of the sampling process. Also, cumulative plots of BIN richness over the season would be very welcome. Additional minor point: log-length should be used if using a log link in the GAM-models.

Comparisons in the discussion to Hallmann et al and Seibold et al are invalid in my opinion. While the authors defend the space-for-time-substitution in the introduction, I find it a stretch to draw

conclusions by comparing the results to these two, or any other longitudinal study.

Minor points

L33 Remove multidiversity.

L60 Urban ecologists may disagree

L82 add substitution after space-for-time

L102 Not clear. Has spatial autocorrelation and location random effect been included in this model? Providing a model equation might help at this point.

Fig1a) are those partial effects? Then maybe seasonal effects BINrichness are masked by strong effect of temperature?

Table 1 The coefficient for Agriculture is positive and significant? I found no discussion about this.

L382 "Orthogonal" How is a flight corridor determined here? And doesn't this artificially inflate trap chances? If so, how do you control for differences in flight corridors (e.g length or number of habitats) at each location?

L387 Ethanol 80%. Maybe problematic for dna preservation if at high temperatures for extensive periods of time (14days here).

L407 Just Reported in the RL? That includes species for which an assessment may indicate favorable conservation status. Don't you want to use the proportions of species in each category?

L432. Log-offset?

L212-214 Your findings do not show any "reductions" in richness or biomass. That is impossible given one year of data. Need rephrasing. As a matter of fact I think all comparisons in this whole paragraph are somewhat beyond what you can say based on this dataset.

L215 followed different.... land-use types? Rephrase

L216-217 This comparison is invalid. Hallmann et al discuss temporal declines, not landuse-variation. They are not reductions. It is equally probable insect biomass and diversity were already much lower in urban vs non-urban settings.

L219-220 Again, you cannot compare magnitudes of decline between this study and the latitudinal study of Seibold et al .

L221-222 Doesn't this theory also predict a hump in richness?

L225 The comparison to opposing species~biomass trends over time is somewhat invalid I think. The sources referred to discuss multiyear assessments, not within year phenological changes.

Reviewer #2 (Remarks to the Author):

The authors report on a regional study of insect biomass and diversity along land-use and elevational

gradients in Bavaria, Germany. The study is framed as a space-for-time approach to understanding insect declines recently reported in long-term studies that typically have been conducted in more limited spatial extents of climatic and land-use variation. Overall, the study design was well planned and replicated to provide good spatial and land-cover representation of the region. Insect sampling involved standard Malaise traps and uses DNA metabarcoding for species identifications, which provided good taxonomic coverage. While there are a large number of regional studies that examine climatic and land-use effects on insect biodiversity, this study does encompass a larger number of taxa than most previous studies. Since this is a one-year study, however, it does not directly address long-term declines in insect biomass and diversity, though there are findings that are relevant to recently published studies on long-term insect declines.

1. Starting at the back of the paper with the methods, I think the authors have done an excellent job in carefully planning the spatial layout of the study landscapes. Through no fault of their own, however, only the urban-dominated landscapes are more highly interspersed across the region, whereas the agricultural and semi-natural landscapes tend to be more co-located in specific areas which may lead to spatial correlation in the insect faunas that are present. This is a common challenge in regional studies of land-use effects, but I'm not sure that their statistical models capture it in the way that it is stated. The random effect of geographic location in their statistical models partly accounts for spatial autocorrelation in landscapes, but without a temporal autocorrelation term in the statistical model to account for repeated measures, the spatial and temporal autocorrelation would seem to be confounded. I understand that biomass and species richness are typically used as response variables in studies of insect decline, but it would be helpful to have some sense of the broader compositional shifts that occur in urban and agricultural landscapes besides those reflected in red-listed species. I did not find the environmental correlation matrix (Fig. 4) particularly helpful; this finding can be reported in a single sentence or put in supplementary materials.

2. In the introduction, the authors state that a space-for-time substitution is the best available design for studying insect declines associated with land-use changes. It is clear that many long-term studies of insect decline are of limited spatial or taxonomic extent, but this statement also makes several assumptions about the past environmental filtering of species pools that are currently available to colonize habitats. Long-term studies suggest that this is indeed the case. So, despite their limitations, long-term studies do provide different insights into insect declines than are possible with a space-for-time study design. The introduction is almost entirely focused on temporal patterns of decline, when the study does not directly measure temporal declines. In contrast, there is only brief mention of the many regional studies that have examined land-use / land-cover effects on insect diversity and composition.

3. I can appreciate the time and expense required for DNA metabarcoding of a large number of samples, but Fig. 1a suggests that the analysis of at least 1-2 additional early season samples are needed to capture the peak insect diversity of the region. I'm also confused about 1a because it seems to plotting the changes in the absolute values of richness and biomass with season but the y-axis indicates that it is the multiplicative effect of season so I assume these are model predictions that are smoothed over the different time intervals? The other panels in Fig. 1 make more sense, but what are the error envelopes? Are they prediction intervals from the random-effects model or confidence intervals of the fixed effects? It makes a big difference. In Fig. 2 each of the local and landscape variables are compared statistically using a very large number of multiple comparisons which I did not find very meaningful. Alternatively, the percent change or standardized effect could

be expressed with a confidence interval and whether it differs from zero. The statistical significance and magnitude of the effect would be clear and would not entail such a large number of pairwise multiple comparisons.

4. In the discussion, the authors compare their results to those of the Hallman et al. (2017) study in terms the seasonal patterns of biomass, which indeed does confirm their findings from a long-term data series. However, rather than stating that biomass cannot be used to predict peak richness (which as noted above was not actually detected in this study), please provide biological explanations for why this is the case. The most logical explanation is that most of the diversity of early season species are small-bodied and those that emerge in later season tend to be more large-bodied species, but again the authors seem to be hung up on the applicability of their findings or even refuting conclusions from studies of long-term declines. The positive effects of higher temperature and precipitation make sense biologically, but over wider climatic gradients we would expect these to be nonlinear as with long-term climate change. There is evidence of this from other diversity studies along elevational gradients. Returning to the co-location of land-use types within the region, the land-use and climatic gradients are not entirely independent and should be mentioned as a study limitation. Nonetheless, the conclusions of the study are still quite sound and solidly based on the findings that landscapes with greater agricultural and urban land uses have lower insect biomass or diversity.

Reviewer #3 (Remarks to the Author):

As per the editor's request, my focus has been on the methodologies and on the data obtained using insect metabarcoding.

In this specific regard, I am very pleased with the presentation of the work "Relationships of insect biomass and richness with land use along a climate gradient".

I have found a number of very good points:

- The authors decided to use a 97% genetic similarity threshold to consider their BINs, which I think is a very smart choice. This allows to proceed with the following analysis without having to further discuss a number of species recorded (which would be challenging). Instead, using the BINs as a proxy for their diversity measurement appears to be the best choice.
- Methodologically, I appreciated the separation between large and small insects, which avoids having some of the samples "flooded" with the reads of the largest specimens.
- The authors used control samples and specified in their methodology how they used these in order to correct for possibly contaminations. It is really nice to finally see papers stating these aspects.

Alas, the funding limitations have sadly precluded what I think would have made this dataset even more exciting. Based on Figure 1a, the part of the data that was analysed for species richness using metabarcoding is the part with the lower richness. By not including the period March-May, where a high number of pollinators would start their activity, the authors have missed what I suspect (based on Figure 1a) would have been their peak in richness. Since metabarcoding analysis of the whole

dataset was not an option, the authors decided to focus on a the period including the end of spring and the beginning of summer. Based on their possibility, I think this was a good choice, allowing to compare the variation happening with the change of the season.

I cannot but hope for a follow-up work including data from the whole year!

I have only two minor comments that can be easily addressed and a few very minor corrections (see below). Other than this, the metabarcoding techniques and the data obtained with them are solid and provide very useful information.

I suggest the manuscript is accepted after very minor revisions.

Comments:

Gene(s) used.

I would like to read in full what genes have been used for this work. I am confident the authors have used fragments of the subunit I of the cytochrome oxidase gene (COI), since they refer to a COI database, and this is the most used marker for insect barcoding and metabarcoding. However, nowhere in the manuscript the full name of the gene appears, not even in the supplementary materials. In order to ascertain the gene used, the reader would have to follow up the trace of reference papers listed, from Hausmann et al. 2020 to the papers of Morinière. This is not ideal and certainly not practical.

I would suggest to include:

- The full name of the gene followed by its abbreviation in the main text.
- The primers names and sequences in the supplementary materials.

This will be extremely handy to anyone trying to use this work as an example for future research.

Samples used:

Pages 18-19: There appears to be a variation between the periodicity of sampling described and the actual number of samples. For example, collecting fortnightly from the beginning of April to mid-August should allow for 9 collections (mid-April, end of April, mid-May, end of May, mid-June, end June, mid-July, end July, mid-August). The authors reported only 8 complete sampling campaigns on all plots, but they state this was variable.

If I understood the number correctly, there were 60 quadrants, each with 3 plots, and each plot was sampled 8 times = 1440 samples. There are 147 samples missing from the insect biomass analysis (~10%). I perfectly understand that this might be due to the variable logistical constraints that did not allow to sample more regularly, but I still think the authors should explain this in detail. Were these 147 samples not sampled at all? Sampled but discarded for some reason? Whoever works with field-collected trap samples is well aware of the wide range of issues that might happen during field work season. As long as the fate of the samples is meticulously reported, and it does not bias the final result, this is not a reason of concern. However, if the authors do not specify this, a suspicious reader (or reviewer!) may wonder if that 10% of samples missing is coming from the same field-use gradient or from the same plot. In this case, the percentage of missing data would affect the results.

The authors should be more specific and detail what samples could be collected. Potentially, a

supplementary table stating the origin of each sample would clarify any doubt. If the table could include origin of the sample (quadrant, plot and time of sampling), and if it was used for metabarcoding, that would be fantastic.

Minor corrections:

Line 349: In order not to create confusion for the reader, I would specify that for each quadrant were chosen the 3 most common land-use types OUT OF A LIST OF FOUR.

So, within the brackets, I suggest adding "Of the four types considered in this study:".

It should read "(Of the four types considered in this study: forest, meadow, arable field and settlement)"

Line 356: missing space between "3x30" and the unit "m", and between "0.5" and the unit "ha". Please, check the whole manuscript and be consistent: either include a space or remove it everywhere.

Line 379: Similarly, missing space between "0.90" and the unit "m".

Line 380: Missing unit "m" after "0.90". Again, for consistency, include the unit everywhere.

Line 390: Add a "to" between "done" and "improve". It should read "This was done to improve barcoding results".

Lines 399-400: I would rephrase the sentence as follows: "The BIN clusters match the actual taxonomically identified species at different levels (between 90% and 99% COI genetic similarity), depending on the taxa." [A good point where the gene can be included].

[Editor's note: Reviewer 3 was also asked to comment on certain points raised by Reviewer 1 due to their technical expertise, see below]

"Investigating /sorting out samples out 179x8 traps is a daunting task at best. Use of DNA metabarcoding as surrogate of manual species identification is state of the art among approaches to tackle this logistical constraint. However, using BINs as a surrogate of species richness does not come without a cost, and I have the feeling that the authors have not appreciated (or warned the reader) enough about risks associated with using BINs as surrogate for richness."

While this might be considered an issue of semantics, I don't think the authors used BINs as a surrogate of richness. They used BINs as a unit to measure richness. As per my comment in the original review, this is actually a smart move. Indeed, the authors are not attempting to state they know the diversity of the area based on BINs, instead, they used BINs as a measurement to determine the species richness. Of course there are other ways to measure richness but using BINs as a unit will generate a perfectly valid results for BINs richness.

In order to meet Reviewer 1 mid-way, the authors should probably state more clearly potential issues linked to the use of BINs and explain why they decided to use this specific measurement for their biodiversity assessment (and, for example, why they didn't use ASV richness).

I am particularly intrigued by the lack of a seasonal BIN-richness pattern (i.e. not peaking along with biomass). This is at odds with what is currently seen by comparison of biomass with manual species richness determination under the same trapping conditions (see Hallmann et al PNAS 2021).

Expecting BINs richness to peak with biomass is a plainly wrong assumption, I am afraid. Biomass could peak if ten particularly large beetles of the same species were to fall in the same trap, but this would still be a single BIN due to the ten beetles being co-specific. Similarly, ten species of thrips

could weigh less than one single beetle's leg.
Biomass and biodiversity are not exchangeable.

"I suspect something is off here. One possibility for example is that DNA has been degraded in the warmer sampling weeks of June-July, affecting the number of BIN-reads. The 80% solution used in this study during trapping is not likely to stay 80% but drop significantly below at hotter days, which undoubtedly affects DNA preservation (see also Marquina et al 2021, PeerJ 9:e10799). At best, this should be discussed in the paper, but I believe it is very worthwhile to investigate this. Eg by measuring ethanol concentration at sample collection date."

Now, on the ethanol concentration in the traps, Reviewer 1 might have a point.

Leaving traps outside for two weeks during the warmer months can surely lead to ethanol evaporation.

However, the authors had been collecting their traps periodically since earlier months. Had they noticed high levels of evaporation, I would expect they would have stated it and, what is more important, they would have acted on it by either changing the solution % or adding more ethanol every few days.

The reason I previously did not comment on this is that the authors did not mention evaporation in their methods, and I (perhaps wrongly) assumed they did not observe it.

To avoid any doubt, the authors could be asked to state something on the lines of "no significant evaporation could be observed/ negligible levels of evaporations were observed" [*Editor's note: if no observations on ethanol levels were made, please acknowledge it*]

"I'm surprised to not see a correlation plot between biomass and number of BINs per sample. Even more interesting would be a temporal (seasonal) correlation plot, which would help understanding the data better in relation to the previous comment."

I think this is a fair request. Indeed, it would be quite interesting to see the plots.

Ideally, it would be nice to observe these plots also separated between the different land-use areas.

"L387 Ethanol 80%. Maybe problematic for dna preservation if at high temperatures for extensive periods of time (14days here)."

Only if this leads to evaporation, that I am aware of.

Point to point:

Please find our answers in bold.

REVIEWER COMMENTS

Reviewer #1 (Remarks to the Author):

The authors present a novel study on variation in insects across a number of land-use categories, and investigate phenological and landscape effects on biomass and species richness, the latter as inferred from DNA metabarcoding. They compare land use and climatic effects at two different spatial and temporal scales. They conclude that urbanization and agriculture are major drivers of decline, increased amount of insects along temperature gradients, and that richness and biomass are not exchangeable metrics. I have a number of concerns and suggestions for improvements.

We thank the reviewer for the constructive comments. Since reviewer 3 was asked to comment on some of the remarks of reviewer 1 (marked in gray below), we reply to those in tandem in the section of reviewer 3.

Concerns:

Investigating /sorting out samples out 179x8 traps is a daunting task at best. Use of DNA metabarcoding as surrogate of manual species identification is state of the art among approaches to tackle this logistical constraint. However, using BINs as a surrogate of species richness does not come without a cost, and I have the feeling that the authors have not appreciated (or warned the reader) enough about risks associated with using BINs as surrogate for richness.

I am particularly intrigued by the lack of a seasonal BIN-richness pattern (i.e. not peaking along with biomass). This is at odds with what is currently seen by comparison of biomass with manual species richness determination under the same trapping conditions (see Hallmann et al PNAS 2021).

See our reply below.

I suspect something is off here. One possibility for example is that DNA has been degraded in the warmer sampling weeks of June-July, affecting the number of BIN-reads. The 80% solution used in this study during trapping is not likely to stay 80% but drop significantly below at hotter days, which undoubtedly affects DNA preservation (see also Marquina et al 2021, PeerJ 9:e10799). At best, this should be discussed in the paper, but I believe it is very worthwhile to investigate this. Eg by measuring ethanol concentration at sample collection date.

I'm surprised to not see a correlation plot between biomass and number of BINs per sample. Even more interesting would be a temporal(seasonal) correlation plot, which would help understanding the data better in relation to the previous comment.

See our reply below.

I am confused about what "red-listed species" mean in this context? Species that made it on the red list? Or that they are under some threat-category? Making it on the list does not translate to unfavorable conservation status, as many species have non-threatened status. So why is the sum of red list species -which is what I assume is what has been used- an

interesting response variable as opposed to the proportions of the categories? Also, there seems to be a mixing between two different red lists, which resemble different spatial scales (state vs country). This is inappropriate.

Thank you for pointing this out. In conservation biology, only species that are also assigned to an endangerment category are referred to as red-listed species. We followed this classification, as only species of the categories critically endangered, endangered, vulnerable, and near threatened were considered as red-listed species in our study. We explain this choice more clearly in the revised manuscript (p. 22, l. 467):

For red-listed species, the number of species reported in one of the categories critically endangered (CR), endangered (EN), vulnerable (VU), or near threatened (NT) as reported in Red Lists for the federal state of Bavaria and for Germany was counted based on a meta-barcoding species identification with > 97% probability.

Regarding the comment on our use of two Red Lists, one for Germany and one for Bavaria, it should be noted that we evaluated a taxonomically unique, comprehensive data set of arthropods from a wide variety of taxa. Unfortunately, there is as yet no Red List, neither for Bavaria nor for Germany, that includes all of these groups. Moreover, since Bavaria covers a considerable part of Germany, the Red List of Germany is also of interest and is regularly used as a supplement in nature conservation assessments conducted by scientists and federal state authorities in Bavaria. Note also that we distinguished only between red-listed and not red-listed species, without considering the threat level. While the threat level may differ between state and national levels, the coarse classification as red-listed is similar at both scales. Our aim was to highlight species of high importance for conservation on the basis of their endangerment, an approach that is not unusual in conservation biology (see, e.g., (Beudert et al. 2015). We added a sentence addressing this point in the text. (p. 22, l. 470):

The consideration of both national and federal scales was necessary to represent the broad taxonomic spectrum of our samples, which is not consistently covered by either of the two Red Lists alone, and was justified by the fact that Bavaria covers ~20% of Germany and includes most landscape types found in the country.

I notice the use of gradients throughout the paper. However I am under the impression from the materials section that you used land use categories. Not gradients.

While we understand your concern, the categories were selected as part of a stratified sampling along a gradient of increasing land-use intensity. At the local habitat scale, this ranged from forests to meadows to arable fields to settlements, and at the landscape scale from near-natural to agricultural to urban landscapes. We revised the text to clarify our approach. (p. 4, l. 92):

Thus, in a space-for-time substitution approach we set up 179 malaise traps in 2019 along a local land-use gradient of increasing intensity, ranging from forests, to meadows, to arable fields, and finally to settlements, thereby including the full range of land-use intensities in temperate Europe²⁴. The level of land-use intensity in the surrounding landscape (ranging from semi-natural to agricultural to urban) was also considered²⁴.

Trapping periods seems to be variable (although not specified how much), roughly two weeks. While averaging quantities over the trapping periods is probably ok for biomass analysis, this is certainly not the case for species richness, and by extension the BINs, because richness does not act additively with increasing survey effort (i.e. variable trapping durations

per sample). As such, I suggest to add survey time per sample as a fixed, possibly smooth effect, and not as an offset in order to have a better description of the sampling process.

Here, we do not agree with the reviewer's interpretation. First, we sought to achieve a consistent trapping duration of 14 days but, due to logistic constraints, the trapping period diverged from this target (average trapping period: 15.23 ± 2.25 days). Second, both biomass and species richness are expected to increase with increasing sampling effort, i.e., longer trapping duration. All of our models regressed the expected response per time unit, which was technically implemented as a $\log(\text{length})$ offset in the Gaussian (biomass) and negative binomial (species richness) models. There was no additivity assumption over the trapping periods, and we modeled a fraction rather than the absolute values, For the importance of accounting for the sampling effort, see (Gotelli and Colwell 2001).

Also, cumulative plots of BIN richness over the season would be very welcome.

We followed this suggestion and in the revised manuscript have provided cumulative plots of BIN richness over the seasons. Because this ignored certain major assumptions in statistics and the core models were applied to the raw data considering all potential constraints of the data, we decided to add this descriptive graph to the Supplementary information (Fig. S3).

Additional minor point: log-length should be used if using a log link in the GAM-models.

Here the reviewer is correct, and we used log-length for all models with a log link, i.e., the Gaussian model and the negative binomial model (default), as described in the Methods section.

Comparisons in the discussion to Hallmann et al and Seibold et al are invalid in my opinion.

While the authors defend the space-for-time-substitution in the introduction, I find it a stretch to draw conclusions by comparing the results to these two, or any other longitudinal study.

We fully agree that space-for-time approaches do not substitute for longitudinal studies. However, the former include assumptions on changes in land use as a major driver, as was the case in Hallmann et al., who considered changes in the surrounding agriculture. Therefore, in our opinion, it is valid to compare the magnitude of the difference in biomass or species number across habitats or landscapes with the reported changes in a habitat over time. For example, if the temporal decline in biomass within small, protected areas surrounded by an agricultural landscape is the result of a landscape-wide decline in the agricultural landscape, then a difference of similar magnitude should be seen in agricultural and semi-natural landscape types. This point is now addressed in a sentence added to the Introduction. (p. 3, l. 75):

Space-for-time studies cannot replace long-term time series, but they are complementary, helping to fill gaps in long-term data series in the short term. Moreover, space-for-time studies allow the inclusion of a large number of sampling locations and therefore assessments of the combined impacts of climate and land-use intensity across all land-use types, from semi-natural to agricultural to urban.

The paragraph in the discussion has been rewritten. It now reads (p. 12, l. 239):

The up to 40% difference in insect biomass between land-use intensities did not reach the magnitude of > 75% reported for temporal decline reported in Hallmann et al.⁶ Moreover, biomass was not lowest in agricultural areas, as discussed by those authors, but in urban habitats. If the temporal decline in biomass in small, protected areas surrounded by an agricultural landscape, such as the experimental sites in Hallmann et al.⁶, was the result of

a landscape-wide decline within the agricultural landscape, then in our study there should have been a difference of similar magnitude between the agricultural and semi-natural landscape types, which was not the case.

Minor

points

L33 Remove multidiversity.

Changed accordingly

L60 Urban ecologists may disagree

Thank you for pointing this out. Our intention was to say that in land-use intensity studies these urban habitats are often ignored (see the major papers discussed in the field of insect decline, as well as the conclusion on research gaps in the most recent global meta-analysis by Van Klink et al 2020), not ignored as such in science. We changed the text accordingly. (p. 3, l. 64):

Moreover, land-use intensity studies have been largely restricted to forests and grasslands¹³ or have focused on specific land-use effects, such as those within agricultural or urban areas^{14,15}

L82 add substitution after space-for-time

Changed accordingly

L102 Not clear. Has spatial autocorrelation and location random effect been included in this model? Providing a model equation might help at this point.

Smooth spatial and temporal effects are present in all models. We provided the complete source code of all analyses, so that interested readers can consult this source for the exact model description and parameterization.

Fig1a) are those partial effects? Then maybe seasonal effects BINrichness are masked by strong effect of temperature?

Yes, the effects are partial and were assumed to act additively in these models. We modified the figure caption accordingly.

Table 1 The coefficient for Agriculture is positive and significant? I found no discussion about this.

This is discussed in the Discussion section, where we explain this effect by higher productivity. The sentence has been rewritten as follows (p. 16, l. 322):

The contrasting responses of biomass and richness point to differences in the respective mechanisms. Insect biomass is positively related to productivity and is thus highest in agricultural landscapes and in forests habitats embedded in agricultural landscapes managed to maximise plant productivity and continuous plant biomass^{46,47}. Insect biomass is lowest in urban environments, where productivity is limited due to a high percentage of sealed areas without vegetation.

L382 “Orthogonal” How is a flight corridor determined here? And doesn’t this artificially inflate trap chances? If so, how do you control for differences in flight corridors (e.g length or number of habitats) at each location?

The sentence in the Methods section has been rewritten as follows (p. 21, l. 433):

At each plot, one malaise trap was installed in the plot center. To avoid restricted access to the traps by small woody features (e.g. shrubs), the traps were oriented orthogonal to the edge of patches with high vegetation. In plots without high vegetation, the traps were

randomly positioned. In addition, vegetation in close proximity to the trap entrance was kept low throughout the season.

L387 Ethanol 80%. Maybe problematic for dna preservation if at high temperatures for extensive periods of time (14days here).

See our reply below.

L407 Just Reported in the RL? That includes species for which an assessment may indicate favorable conservation status. Don't you want to use the proportions of species in each category?

Again, thank you for pointing this out. We rewrote a subsection within Methods to clarify our approach. See the above comment.

L432. Log-offset?

Yes; in the negative binominal model log-link is the default.

L212-214 Your findings do not show any "reductions" in richness or biomass. That is impossible given one year of data. Need rephrasing. As a matter of fact I think all comparisons in this whole paragraph are somewhat beyond what you can say based on this dataset.

To avoid a misunderstanding, the text has been changed. (p. 12, l. 235)

We found the lowest species richness in arable fields embedded in agricultural landscapes, and the lowest biomass in urban landscapes.

L215 followed different.... land-use types? Rephrase

Changed to land-use intensities.

L216-217 This comparison is invalid. Hallmann et al discuss temporal declines, not landuse-variation. They are not reductions. It is equally probable insect biomass and diversity were already much lower in urban vs non-urban settings.

We rewrote this paragraph and no longer use the term "reductions." However, we believe our comparisons to be valid, since Hallmann et al. in fact discuss land-use variations. Also see the comment above.

L219-220 Again, you cannot compare magnitudes of decline between this study and the latitudinal study of Seibold et al .

We now make clear that our study complements those time-series studies. See the comment above.

L221-222 Doesn't this theory also predict a hump in richness?

We removed this reference.

L225 The comparison to opposing species~biomass trends over time is somewhat invalid I think. The sources referred to discuss multiyear assessments, not within year phenological changes.

We agree. When referring to Hallman et al. our intention was to focus on the difference in land-use response, but the temporal divergence is also of importance due to the nature of the different approaches and to sampling during different seasons.

Reviewer #2 (Remarks to the Author):

The authors report on a regional study of insect biomass and diversity along land-use and elevational gradients in Bavaria, Germany. The study is framed as a space-for-time approach to understanding insect declines recently reported in long-term studies that typically have been conducted in more limited spatial extents of climatic and land-use variation. Overall, the study design was well planned and replicated to provide good spatial and land-cover representation of the region. Insect sampling involved standard Malaise traps and uses DNA metabarcoding for species identifications, which provided good taxonomic coverage. While there are a large number of regional studies that examine climatic and land-use effects on insect biodiversity, this study does encompass a larger number of taxa than most previous studies. Since this is a one-year study, however, it does not directly address long-term declines in insect biomass and diversity, though there are findings that are relevant to recently published studies on long-term insect declines.

1. Starting at the back of the paper with the methods, I think the authors have done an excellent job in carefully planning the spatial layout of the study landscapes. Through no fault of their own, however, only the urban-dominated landscapes are more highly interspersed across the region, whereas the agricultural and semi-natural landscapes tend to be more co-located in specific areas which may lead to spatially correlation in the insect faunas that are present. This is a common challenge in regional studies of land-use effects, but I'm not sure that their statistical models capture it in the way that it is stated. The random effect of geographic location in their statistical models partly accounts for spatial autocorrelation in landscapes, but without a temporal autocorrelation term in the statistical model to account for repeated measures, the spatial and temporal autocorrelation would seem to be confounded. **Thank you for your comment. A smooth temporal term is included in all models and, in addition to the smooth spatial term, handles spatio-temporal autocorrelation in the models.**

I understand that biomass and species richness are typically used as response variables in studies of insect decline, but it would be helpful to have some sense of the broader compositional shifts that occur in urban and agricultural landscapes besides those reflected in red-listed species.

We agree that species composition is a highly interesting target variable and plan to present the composition results in a second paper, combined with plant survey data. In this paper, our focus was on the variables predominantly discussed in connection with insect decline, i.e., richness and biomass. Including species composition as well as a more finely resolved evaluation, e.g., at the family level, would have been too broad-ranging for one manuscript and would have hindered the presentation and discussion of the results in sufficient detail.

I did not find the environmental correlation matrix (Fig. 4) particularly helpful; this finding can be reported in a single sentence or put in supplementary materials.

According to your suggestion, we removed the graph and added a sentence (p. 23, l. 484).

The correlation between climate variables was low for most variables (Pearson's $r < 0.23$), and only moderate for the long-term mean annual average near-surface temperature and long-term mean annual precipitation (Pearson's $r = 0.51$).

2. In the introduction, the authors state that a space-for-time substitution is the best available design for studying insect declines associated with land-use changes.

We apologize for the misunderstanding. We meant that the available long-term data contain considerable gaps with regard to land-use intensities (especially high-intensity areas; see van Klink et al. 2020 *Science*) and different species groups. It will take many years before comprehensive long-term data become available. In the meantime, space-for-time studies are the best approach available to investigate still-open questions about the drivers of species decline in a timely manner. See the comments above.

It is clear that many long-term studies of insect decline are of limited spatial or taxonomic extent, but this statement also makes several assumptions about the past environmental filtering of species pools that are currently available to colonize habitats. Long-term studies suggest that this is indeed the case. So, despite their limitations, long-term studies do provide different insights into insect declines than are possible with a space-for-time study design. The introduction is almost entirely focused on temporal patterns of decline, when the study does not directly measure temporal declines. In contrast, there is only brief mention of the many regional studies that have examined land-use / land-cover effects on insect diversity and composition.

We fully agree with your point and have added a statement that space-for-time studies are an important supplement to existing long-term data, as they are able to close knowledge gaps in the short term (p. 3, l.75).

Space-for-time studies cannot replace long-term time series, but they are complementary, helping to fill gaps in long-term data series in the short term. Moreover, space-for-time studies allow the inclusion of a large number of sampling locations and therefore assessments of the combined impacts of climate and land-use intensity across all land-use types, from semi-natural to agricultural to urban.

We also revised the text to point out other studies on specific land-use effects (p. 3, l. 64): *Moreover, land-use intensity studies have been largely restricted to forests and grasslands¹³ or have focused on specific land-use effects, such as those within agricultural or urban areas^{14,15}*

3. I can appreciate the time and expense required for DNA metabarcoding of a large number of samples, but Fig. 1a suggests that the analysis of at least 1-2 additional early season samples are needed to capture the peak insect diversity of the region.

As always, research funds are limited. We therefore deliberately chose key time periods, i.e., those for which a complete set of samples was available. Since in some study sites there was still snow in April and the first half of May was very cold and very rainy, metabarcoding was conducted beginning with the samples from the second half of May. At the time of this writing, we are applying for additional funding to sequence more samples. However, we do not expect that further sequencing data will change our main findings.

I'm also confused about 1a because it seems to plotting the changes in the absolute values of richness and biomass with season but the y-axis indicates that it is the multiplicative effect of season so I assume these are model predictions that are smoothed over the different time intervals?

The plot presents the partial multiplicative effect of season, which is a smooth term acting multiplicatively on the expected outcome per time unit. We added this information to the figure caption.

The other panels in Fig. 1 make more sense, but what are the error envelopes? Are they prediction intervals from the random-effects model or confidence intervals of the fixed effects? It makes a big difference.

Error envelopes depict standard errors below and above the estimated mean responses. We added this information as well.

In Fig. 2 each of the local and landscape variables are compared statistically using a very large number of multiple comparisons which I did not find very meaningful. Alternatively, the percent change or standardized effect could be expressed with a confidence interval and whether it differs from zero. The statistical significance and magnitude of the effect would be clear and would not entail such a large number of pairwise multiple comparisons.

We deliberately chose this approach to allow statistically valid comparisons between categories, using the correct post-hoc test. This is the only way to determine whether a category A is better than a category B. The choice of model also clearly showed the absolute differences in biomass or species numbers between categories. Since a wide variety of these comparisons are likely to be of interest in political decision-making processes, we tested all multiple comparisons. Our large sample size enabled the use of statistical methods (see annotated R code). Nevertheless, to simplify interpretations of the results, the figure presents the most important differences. As the reader has access to all of the raw data and the R code, other details can be reproduced and extracted at any time.

4. In the discussion, the authors compare their results to those of the Hallman et al. (2017) study in terms the seasonal patterns of biomass, which indeed does confirm their findings from a long-term data series. However, rather than stating that biomass cannot be used to predict peak richness (which as noted above was not actually detected in this study), please provide biological explanations for why this is the case. The most logical explanation is that most of the diversity of early season species are small-bodied and those that emerge in later season tend to be more large-bodied species, but again the authors seem to be hung up on the applicability of their findings or even refuting conclusions from studies of long-term declines. **Thank you for the comment. Similar remarks were made by reviewers 1 and 3. We therefore added new analyses showing that the biomass does not purely follow the richness of the two species-richest groups but rather the richness of the two groups with many large species (Orthoptera and Lepidoptera). This result supports the interpretation of reviewer 2, that the summer peak in biomass was driven by certain, mainly large-bodied taxa that are, however, not extremely rich in species. This information is presented in the new Figure 3 and discussed in a new paragraph in the Discussion (p. 15, l. 312):**

Across all habitats, biomass was best explained by the increase in BIN richness of three species groups, Orthoptera, Lepidoptera and Diptera. Of the diverse taxa Hymenoptera and Diptera, only the richness of the latter positively affected total biomass, and it was principally the richness of the two groups with many large species (Orthoptera and Lepidoptera) driving the seasonal effect. This can be explained by the fact that Lepidoptera abundance peaks in July⁴⁴, thus coinciding with the higher abundances of most species of hemimetabolous Orthoptera during the summer⁴⁵, and therefore well accounting for the biomass peak in summer. This finding also suggests that the different patterns of biomass and total species numbers primarily derive from shifts in species composition, which shows

the importance of taxon comprehensive studies in investigations of biomass and species richness.

The positive effects of higher temperature and precipitation make sense biologically, but over wider climatic gradients we would expect these to be nonlinear as with long-term climate change. There is evidence of this from other diversity studies along elevational gradients. Returning to the co-location of land-use types within the region, the land-use and climatic gradients are not entirely independent and should be mentioned as a study limitation.

We agree and have modified the Discussion to clarify this point. The relevant sentence now reads (p. 18, l. 378):

However, further warming, extreme heat, and drought events may negatively affect biodiversity, although non-linear responses can be expected in other climates or across longer gradients.

Nonetheless, the conclusions of the study are still quite sound and solidly based on the findings that landscapes with greater agricultural and urban land uses have lower insect biomass or diversity.

Thank you for your remarks and positive evaluation.

Reviewer #3 (Remarks to the Author):

As per the editor's request, my focus has been on the methodologies and on the data obtained using insect metabarcoding.

In this specific regard, I am very pleased with the presentation of the work "Relationships of insect biomass and richness with land use along a climate gradient".

Thank you for the positive evaluation.

I have found a number of very good points:

- The authors decided to use a 97% genetic similarity threshold to consider their BINs, which I think is a very smart choice. This allows to proceed with the following analysis without having to further discuss a number of species recorded (which would be challenging). Instead, using the BINs as a proxy for their diversity measurement appears to be the best choice.

Thank you for your feedback. We revised the Methods section to clarify this point for a broader audience (see the comment of reviewer 1).

- Methodologically, I appreciated the separation between large and small insects, which avoids having some of the samples "flooded" with the reads of the largest specimens.

Thank you for your comment. This point is now considered in the Discussion (p. 15, l. 304).

A second methodological reason for the lower BIN richness is that small species are often "overlooked" in very biomass-rich samples⁴¹⁻⁴³. To avoid this problem, we divided each sample into two fractions (small and large species) and sequenced them separately.

- The authors used control samples and specified in their methodology how they used these in order to correct for possibly contaminations. It is really nice to finally see papers stating these aspects.

We thank you for the positive evaluation of our data-processing strategy.

Alas, the funding limitations have sadly precluded what I think would have made this dataset even more exciting. Based on Figure 1a, the part of the data that was analysed for species richness using metabarcoding is the part with the lower richness. By not including the period

March-May, where a high number of pollinators would start their activity, the authors have missed what I suspect (based on Figure 1a) would have been their peak in richness. Since metabarcoding analysis of the whole dataset was not an option, the authors decided to focus on a the period including the end of spring and the beginning of summer. Based on their possibility, I think this was a good choice, allowing to compare the variation happening with the change of the season.

Thank you for your remarks and positive evaluation.

I cannot but hope for a follow-up work including data from the whole year! I have only two minor comments that can be easily addressed and a few very minor corrections (see below). Other than this, the metabarcoding techniques and the data obtained with them are solid and provide very useful information.

We thank you for the positive evaluation. We are seeking additional funding for further rounds of sequencing.

I suggest the manuscript is accepted after very minor revisions.

Thank you for your evaluation of our study in this still rather new field of large-sample and taxonomic diversity evaluation by metabarcoding.

Comments:

Gene(s) used.

I would like to read in full what genes have been used for this work. I am confident the authors have used fragments of the subunit I of the cytochrome oxidase gene (COI), since they refer to a COI database, and this is the most used marker for insect barcoding and metabarcoding. However, nowhere in the manuscript the full name of the gene appears, not even in the supplementary materials. In order to ascertain the gene used, the reader would have to follow up the trace of reference papers listed, from Hausmann et al. 2020 to the papers of Morinière. This is not ideal and certainly not practical.

I would suggest to include:

- The full name of the gene followed by its abbreviation in the main text.
- The primers names and sequences in the supplementary materials.

We added gene abbreviations to the Methods section and primer names and sequences to the Supplementary information.

This will be extremely handy to anyone trying to use this work as an example for future research.

Samples used:

Pages 18-19: There appears to be a variation between the periodicity of sampling described and the actual number of samples. For example, collecting fortnightly from the beginning of April to mid-August should allow for 9 collections (mid-April, end of April, mid-May, end of May, mid-June, end June, mid-July, end July, mid-August). The authors reported only 8 complete sampling campaigns on all plots, but they state this was variable.

Trapping started late on several plots due to late snow and to delays in getting permissions from landowners. Therefore, although trap set-up started in early April,

sampling did not commence until mid-April. This resulted in eight collections per plot. The text has been changed accordingly (p. 21, l. 441).

The traps were activated mid-April and emptied every 2 weeks until mid-August, for a total of eight complete sampling campaigns on all plots. Due to logistical constraints, the individual sampling period was variable.

If I understood the number correctly, there were 60 quadrants, each with 3 plots, and each plot was sampled 8 times = 1440 samples. There are 147 samples missing from the insect biomass analysis (~10%). I perfectly understand that this might be due to the variable logistical constraints that did not allow to sample more regularly, but I still think the authors should explain this in detail. Were these 147 samples not sampled at all? Sampled but discarded for some reason? Whoever works with field-collected trap samples is well aware of the wide range of issues that might happen during field work season. As long as the fate of the samples is meticulously reported, and it does not bias the final result, this is not a reason of concern. However, if the authors do not specify this, a suspicious reader (or reviewer!) may wonder if that 10% of samples missing is coming from the same field-use gradient or from the same plot. In this case, the percentage of missing data would affect the results.

The authors should be more specific and detail what samples could be collected. Potentially, a supplementary table stating the origin of each sample would clarify any doubt. If the table could include origin of the sample (quadrant, plot and time of sampling), and if it was used for metabarcoding, that would be fantastic.

Thank you for your comment. Overall, we included all samples in the original data table, marking those excluded due to technical problems. In total, 179 plots were established since permission for one plot was not granted. Of 1432 possible samples, 93 could not be obtained due to the late start of sampling, collapsed traps, vandalism, or destruction of the samples during transport. Three additional samples were destroyed during handling, resulting in biomass data but no further BIN data. Missing climate data (failed dataloggers) led to the exclusion of 46 additional samples from the statistical analysis. Thus, 139 samples were excluded from the analysis of insect biomass and 27 from the analysis of BINs. For clarity, the corresponding section in the Methods has been revised accordingly. By reporting the distribution of samples that had to be removed across categories, we clearly show the random character of those samples (see Table S2).

Minor corrections:

Line 349: In order not to create confusion for the reader, I would specify that for each quadrant were chosen the 3 most common land-use types OUT OF A LIST OF FOUR. So, within the brackets, I suggest adding “Of the four types considered in this study:”. It should read “(Of the four types considered in this study: forest, meadow, arable field and settlement)”

Changed accordingly

Line 356: missing space between “3x30” and the unit “m”, and between “0.5” and the unit “ha”. Please, check the whole manuscript and be consistent: either include a space or remove it everywhere.

Changed accordingly

Line 379: Similarly, missing space between “0.90” and the unit “m”.

Corrected

Line 380: Missing unit “m” after “0.90”. Again, for consistency, include the unit everywhere.

Corrected

Line 390: Add a “to” between “done” and “improve”. It should read “This was done to improve barcoding results”.

Corrected

Lines 399-400: I would rephrase the sentence as follows: “The BIN clusters match the actual taxonomically identified species at different levels (between 90% and 99% COI genetic similarity), depending on the taxa.” [A good point where the gene can be included].

Changed accordingly

[Editor's note: Reviewer 3 was also asked to comment on certain points raised by Reviewer 1 due to their technical expertise, see below]

“Investigating /sorting out samples out 179x8 traps is a daunting task at best. Use of DNA metabarcoding as surrogate of manual species identification is state of the art among approaches to tackle this logistical constraint. However, using BINs as a surrogate of species richness does not come without a cost, and I have the feeling that the authors have not appreciated (or warned the reader) enough about risks associated with using BINs as surrogate for richness.”

While this might be considered an issue of semantics, I don't think the authors used BINs as a surrogate of richness. They used BINs as a unit to measure richness. As per my comment in the original review, this is actually a smart move. Indeed, the authors are not attempting to state they know the diversity of the area based on BINs, instead, they used BINs as a measurement to determine the species richness. Of course there are other ways to measure richness but using BINs as a unit will generate a perfectly valid results for BINs richness. In order to meet Reviewer 1 mid-way, the authors should probably state more clearly potential issues linked to the use of BINs and explain why they decided to use this specific measurement for their biodiversity assessment (and, for example, why they didn't use ASV richness).

We thank reviewers 1 and 3 for their comments which helped us to clarify our approach. The simple reason underlying our choice of BINs to measure richness was that in some genera, particularly those of Orthoptera, many OTUs occur per BIN or species. Consequently, the BIN concept is closer to the concept of taxonomic richness, whereas richness would have been potentially inflated by the use of OTUs. Findings based on BINs are thus closer to those that would be obtained if the species had been identified morphologically and they cover a broad range of lineages. The text has been rewritten to emphasize this point and a reference was added. (p. 22, l. 459)

To cover the same level of identification for all taxonomic groups, BINs instead of OTUs were used to measure richness, since the latter tends to overestimate richness in some orders, including Orthoptera. The utility of BINs in characterizing formal genetic units independently of an existing classification has been demonstrated⁶⁰. The BIN system of the online platform BOLD is based on a chain of algorithms clustering similar barcode sequences and checking their taxonomic integrity. The BIN clusters match the actual taxonomically identified species at different levels (90–99% COI genetic similarity), depending on the taxa, thus allowing comparisons with studies based on morphological determination.

I am particularly intrigued by the lack of a seasonal BIN-richness pattern (i.e. not peaking

along with biomass). This is at odds with what is currently seen by comparison of biomass with manual species richness determination under the same trapping conditions (see Hallmann et al PNAS 2021). Expecting BINs richness to peak with biomass is a plainly wrong assumption, I am afraid. Biomass could peak if ten particularly large beetles of the same species were to fall in the same trap, but this would still be a single BIN due to the ten beetles being co-specific. Similarly, ten species of thrips could weigh less than one single beetle's leg. Biomass and biodiversity are not exchangeable.

Thank you for the comment. In fact, we now show with additional analyses, that it is only in part the richness of the species-rich Dipteran group that correlates with total biomass and rather the richness of species groups whose members have a generally larger body size, i.e., Orthoptera and Lepidoptera. The former is particularly poor in species (new Fig. 3). Originally, we planned to present this further group-specific analysis in a second paper, but to show that our findings are not methodological artifacts we added the results of those analyses to the revised manuscript. More detailed analyses covering the family level and community composition will be presented in a follow-up paper. Our findings do not conflict with those of the recent study by Hallmann et al., because the family Syrphidae also positively correlated with total biomass. This information will be presented in detail in the second manuscript.

“I suspect something is off here. One possibility for example is that DNA has been degraded in the warmer sampling weeks of June-July, affecting the number of BIN-reads. The 80% solution used in this study during trapping is not likely to stay 80% but drop significantly below at hotter days, which undoubtedly affects DNA preservation (see also Marquina et al 2021, PeerJ 9:e10799). At best, this should be discussed in the paper, but I believe it is very worthwhile to investigate this. Eg by measuring ethanol concentration at sample collection date.”

Now, on the ethanol concentration in the traps, Reviewer 1 might have a point. Leaving traps outside for two weeks during the warmer months can surely lead to ethanol evaporation. However, the authors had been collecting their traps periodically since earlier months. Had they noticed high levels of evaporation, I would expect they would have stated it and, what is more important, they would have acted on it by either changing the solution % or adding more ethanol every few days. The reason I previously did not comment on this is that the authors did not mention evaporation in their methods, and I (perhaps wrongly) assumed they did not observe it. To avoid any doubt, the authors could be asked to state something on the lines of “no significant evaporation could be observed/ negligible levels of evaporations were observed” [Editor's note: if no observations on ethanol levels were made, please acknowledge it]

Again, we thank both reviewers and the editor for the helpful comments. The size of the collection bottles and the rather short sampling periods ensured that sufficient amounts of ethanol were always present in the bottles. This point is now addressed at length in the Discussion section, which considers three reasons, two methodological and one biological, for the poor correlation between biomass and species number. These arguments and the new analytical results support a biological rather than a methodological explanation. (p. 15, l. 299)

A possible methodological reason for the lower BIN richness during the peak of biomass is that high insect biomass occurs during periods of high temperatures, which would have increased evaporation of the ethanol used for preservation, accelerating the degradation of DNA. Similar effects were shown for samples stored over long periods⁴⁰ of time. In our study, however, the collection bottles contained sufficient amounts of ethanol such that a methodological effect due to ethanol evaporation was unlikely.

“I’m surprised to not see a correlation plot between biomass and number of BINs per sample. Even more interesting would be a temporal(seasonal) correlation plot, which would help understanding the data better in relation to the previous comment.”

I think this is a fair request. Indeed, it would be quite interesting to see the plots.

Ideally, it would be nice to observe these plots also separated between the different land-use areas.

We followed your advice, adding correlation plots for the three sampling campaigns and accumulative plots for the four habitats (Fig. S3)

“L387 Ethanol 80%. Maybe problematic for dna preservation if at high temperatures for extensive periods of time (14days here).”

Only if this leads to evaporation, that I am aware of.

We considered this point in the revised Discussion. See our reply above.

References

Beudert, B., C. Bäessler, S. Thorn, R. Noss, B. Schröder, H. Dieffenbach-Fries, N. Foullois, and J. Müller. 2015. Bark beetles increase biodiversity while maintaining drinking water quality. *Conservation Letters* **8**:272-281.

Gotelli, N. J., and R. K. Colwell. 2001. Quantifying biodiversity: procedures and pitfalls in the measurement and comparison of species richness. *Ecology Letters* **4**: 379-391.

Reviewer #1 (Remarks to the Author):

Second review "Relationship of insect biomass and richness with land use along a climate gradient" by Uhler et.al.

The authors have done a great job clarifying matters in their rebuttal and revision. Quite a few of my concerns are alleviated (with respect to Red List comment, and particularly with clarifying that the reader is looking at partial effects, rather than integrated seasonal effects). However, this changes completely the interpretation of the results in my view, and I suggest another revision to be made.

I was very happy to see the additional analysis done with biomass~taxonomic group relationship, and I understand this is (very interesting!) matter for follow up work. Frankly, I think this subsequent taxonomic analysis is best left for a follow up paper, rather than only partially include it here. Instead, I suggest analysing the relationship between BIN-richness (as response) and biomass (as explanatory) given plot and landscape types. I have included a figure (see annexed figure 1) as an example of what I mean.

Further more, I am now convinced there IS a seasonal hump in richness
L131-135

"Both the total richness and the richness of red-listed species showed a convex response to season, with the highest values occurring at the beginning of the sampling period (May) and the lowest values around early July (Fig. 1)"

It is the residual effect (or partial) what is being discussed here and depicted in fig1a, not the total seasonal projection. I think therefore that richness does follow a humped pattern. To assess the seasonal pattern you need to include (at least) temperature and humidity in the projections. I elaborate: For BIN richness metrics, the multiplicative effects show a 5-fold increase with temperature while only a 2-fold decrease with day-number (so still a hump, given the strong rise in temperature, from May (about 100C) to July (about 200C)). In your biomass model, the hump is not so well explained by temperature alone, so the smooth temporal component takes this additional part. In Bin richness models however, temperature explains most of your variation in your model – predictions (see annexed figure 2). Both plots of the raw data as well as the full model predictions (integrated) clearly show hump in richness.

As such, I believe that the following lines:

L240 "Biomass and richness measures followed different Page temporal patterns []"

L254:256 "However, the contrasting phenological patterns of biomass and total species richness were a first proof that these two facets of biodiversity are not generally correlated and therefore cannot be used interchangeably as proxies."

L323-324 "...different patterns of biomass and total species ..."

cannot be justified based on the present data or analysis. I strongly suggest the authors to remove/change these interpretations.

L255 are not generally correlated.

I added a plot based on your data (figure 1). To me, it seems they are quite often correlated, with the exception of meadows (grasshoppers effect?).

A further issue,
L244-248

"If the temporal decline in biomass [...] was the result of a landscape-wide decline within the agricultural landscape, then in our study there should have been a difference of similar magnitude between the agricultural and semi natural landscape types, which was not the case."

I am still confused with this expectation. So what is your assumption here? No landscape-wide decline but only in agricultural landscape types? A landscape wide effect not limited to agriculture? I certainly do not expect an a-priori 75% difference in biomass between the landscape types here. Either biomass-change rates are different between these landscapes, either initial conditions are different, or both are. Who is to say? A 40% difference as has been found here can have come about from any combination of initial conditions and rates of decline. But maybe I am missing the point?

Finally, I appreciate the part in the discussion about ethanol concentration and potential effects on BIN richness, but am not yet convinced about how much did or didn't evaporate over roughly two weeks, and how this may have affected number of BIN reads. I would suggest to defend this by acknowledging the issue, but assume little effect because metabarcoding-analysis was performed on samples mostly June-July (of comparable temperature ranges), this cannot be expected to have affected the samples to a large extent(???).

L315-326 I am particularly happy about this paragraph now.

[Editor's note: see also the attached figures]

Reviewer #2 (Remarks to the Author):

I have no further comments on the revised manuscript.

Reviewer #3 (Remarks to the Author):

I am satisfied with the revised version of this manuscript.

I have a few extra minor comments that I hope will be helpful to the authors. In particular, a reminder to always clearly state (especially in the discussion) what their values represent. When discussing "richness" and "biomass" it is always better to be as clear as possible (also considering some of the reviewers' comments). I highlighted a couple points (see lines 312, 322, 336), but the authors should double-check again the whole manuscript.

Minor edits:

Lines 233-235: I would suggest changing "goes beyond those that measure biomass or assess single taxa to reveal drivers of insect communities" with something that highlights the advantages of the present study instead of pointing against other works (that "those"). Something on the lines of:

"Our approach provides novel data on species richness across independent gradients of land-use intensity and climate. Furthermore, by combining malaise traps and DNA-metabarcoding, our work is not limited to single factors such as biomass measurements or assessment of single taxa to reveal drivers of insect communities."

Lines 240-241: I would remove "Hallmann et al." and just use "elsewhere".

Lines 242-247: As I suggested above, I would recommend not using this kind of comparison, which may be perceived as stating other studies are "wrong". The authors should focus on the perks of their study, and how that changes from others. The authors should rephrase this sentence stating what they found (or did not found) and THEN compare that with the results showed by other studies.

For example: "Our study recorded a temporal decline in biomass between the agricultural and semi-natural landscape types of only XXX. This appears to be in contrast with the results recorded in a similar analysis (Hallmann et al.) which showed a temporal decline in biomass of YYY in small, protected areas surrounded by an agricultural landscape. On the other hand, the variation in total species richness, matched the magnitude of the temporal decline (~35%) determined over a decade in grasslands and forests by Seibold et al.¹³"

This kind of rephrasing shift the focus from other studies to your own (which is more important!).

Line 306: remove "very".

Line 312: The authors should remember to specify which "biomass" or "richness" measurement they used and state it every time they want to discuss it. If you are referring to biomass variation across different areas, then you should specify it. If you have measured BIN richness, then you should specify it.

For example, "Across all habitats, biomass was best explained by.." with "Across all habitats, biomass variation(?) was best explained by.."

Same at line 322, 336 for biomass and richness. Should read "biomass variation" and "BIN richness". Please, check the entire manuscript.

Line 347: "Country-wide strategy"

Line 365: Reword to: "Nevertheless, additional studies should focus on biomes other than the cultivated landscapes of the temperate zone, such as cold boreal, dry Mediterranean, or hot tropical areas. Here, the different characteristics of the biome may result in land-use intensification being of less importance than climate change."

Point to point

Please find our answers in bold. All changes in the text are highlighted in red.

Reviewer #1 (Remarks to the Author):

Second review “Relationship of insect biomass and richness with land use along a climate gradient” by Uhler et.al.

The authors have done a great job clarifying matters in their rebuttal and revision. Quite a few of my concerns are alleviated (with respect to Red List comment, and particularly with clarifying that the reader is looking at partial effects, rather than integrated seasonal effects). However, this changes completely the interpretation of the results in my view, and I suggest another revision to be made.

I was very happy to see the additional analysis done with biomass~taxonomic group relationship, and I understand this is (very interesting!) matter for follow up work. Frankly, I think this subsequent taxonomic analysis is best left for a follow up paper, rather than only partially include it here. Instead, I suggest analysing the relationship between BIN-richness (as response) and biomass (as explanatory) given plot and landscape types. I have included a figure (see annexed figure 1) as an example of what I mean.

Further more, I am now convinced there IS a seasonal hump in richness

L131-135

“Both the total richness and the richness of red-listed species showed a convex response to season, with the highest values occurring at the beginning of the sampling period (May) and the lowest values around early July (Fig. 1)”

It is the residual effect (or partial) what is being discussed here and depicted in fig1a, not the total seasonal projection. I think therefore that richness does follow a humped pattern. To assess the seasonal pattern you need to include (at least) temperature and humidity in the projections. I elaborate: For BIN richness metrics, the multiplicative effects show a 5-fold increase with temperature while only a 2-fold decrease with day-number (so still a hump, given the strong rise in temperature, from May (about 100C) to July (about 200C)). In your biomass model, the hump is not so well explained by temperature alone, so the smooth temporal component takes this additional part. In Bin richness models however, temperature explains most of your variation in your model – predictions (see annexed figure 2). Both plots of the raw data as well as the full model predictions (integrated) clearly show hump in richness.

We thank you for thoughtful critique, including the examination of our raw data. At this point, we strongly agree with the reviewer that our presentation of the partial effects, in particular of season and local temperature, could be misleading to the reader. In order to prevent this, we have once again made it clear in the results that we are dealing with partial effects and added a graph to the supplement (Fig. S2). We also put less emphasis on the comparison of these effects in the discussion, as this was more of a control variable for our study and less of a major finding. We added the following sentence to the results (p. 6, l. 135):

“It is important to note that we separated the change over the growing season into the pure partial effects of season and local temperature and humidity. When local temperature and

humidity were excluded from the models, both species richness and biomass followed a hump-shaped curve, even though the shape was a lot less pronounced for species richness compared to insect biomass (Fig. S2)”

Regarding the correlation of richness and biomass, we prefer to keep the new taxa specific analyses, since it was highly appreciated by our co-authors and helped clarify some of the issues brought up in the first review. For the context dependent correlation of biomass and total richness we currently are working on further analyses with new statistical methods. Because these methods are far from simple, we plan to write a separate paper on this topic. To include these analyses in the existing manuscript would go beyond the scope of our paper.

As such, I believe that the following lines:

L240 “Biomass and richness measures followed different Page temporal patterns []”

To avoid confusion, we removed this phrase.

L254:256 “However, the contrasting phenological patterns of biomass and total species richness were a first proof that these two facets of biodiversity are not generally correlated and therefore cannot be used interchangeably as proxies.”

Here we clarified the partial effect and rephrased the second half of the sentence to avoid any misinterpretation. It now reads (p. 13, l. 255):

“However, the contrasting phenological patterns of biomass and total BIN richness after controlling for temperature, were a first proof that both facets of biodiversity might respond differently, with biomass more strongly driven by season and BIN richness more dependent on local temperature.”

L323-324 “...different patterns of biomass and total species ...”

cannot be justified based on the present data or analysis. I strongly suggest the authors to remove/change these interpretations.

We followed the suggestion and removed the whole sentence.

L255 are not generally correlated.

I added a plot based on your data (figure 1). To me, it seems they are quite often correlated, with the exception of meadows (grasshoppers effect?).

We removed this phrase to avoid misinterpretation. Nevertheless, the correlation plots created by reviewer 1 also show quite a lot of variation. As mentioned above, this is something we are currently working on as part of a separate paper.

A further issue,

L244-248

“If the temporal decline in biomass [...] was the result of a landscape-wide decline within the agricultural landscape, then in our study there should have been a difference of similar magnitude between the agricultural and semi natural landscape types, which was not the case.”

I am still confused with this expectation. So what is your assumption here? No landscape-

wide decline but only in agricultural landscape types? A landscape wide effect not limited to agriculture? I certainly do not expect an a-priori 75% difference in biomass between the landscape types here. Either biomass-change rates are different between these landscapes, either initial conditions are different, or both are. Who is to say? A 40% difference as has been found here can have come about from any combination of initial conditions and rates of decline. But maybe I am missing the point?

Thank you for pointing this out. To clarify we changed the wording and rephrased the sentence in line with the suggestion of reviewer 3. It now reads (p. 13, l. 244):

“Our study recorded a decline in insect biomass of 40% from semi-natural to urban environments, but no decline from semi-natural to agricultural environments. This appears to be in contrast with the results documented in a similar analysis⁶, which showed a temporal decline in insect biomass of > 75% in small, protected areas surrounded by an agricultural landscape. Interestingly, in Hallmann et al.⁶, the few plots in semi-natural landscapes also showed a similar temporal decline as those in agricultural landscapes (Fig. S3b). On the other hand, the variation in total BIN richness matched the magnitude of the temporal decline (~35%) determined over a decade in grasslands and forests by Seibold et al.¹³ “

Finally, I appreciate the part in the discussion about ethanol concentration and potential effects on BIN richness, but am not yet convinced about how much did or didn't evaporate over roughly two weeks, and how this may have affected number of BIN reads. I would suggest to defend this by acknowledging the issue, but assume little effect because metabarcoding-analysis was performed on samples mostly June-July (of comparable temperature ranges), this cannot be expected to have affected the samples to a large extent(??).

Thank you for your comment. However, our metabarcoding was applied to samples from May, June and July, so it is difficult to completely follow your argument. We believe the stronger argument is that we couldn't observe any dried-out bottles. We further discussed this issue with one of our co-authors (Jerome Moriniere). He is handling several thousand of samples from Malaise traps from Central Europe annually and evaporation has not been a major issue when metabarcoding samples from Malaise traps that were emptied twice per month.

L315-326 I am particularly happy about this paragraph now.

[Editor's note: see also the attached figures]

We appreciate the feedback!

Reviewer #2 (Remarks to the Author):

I have no further comments on the revised manuscript.

Thank you for the positive reply.

Reviewer #3 (Remarks to the Author):

I am satisfied with the revised version of this manuscript.

I have a few extra minor comments that I hope will be helpful to the authors. In particular, a reminder to always clearly state (especially in the discussion) what their values represent. When discussing "richness" and "biomass" it is always better to be as clear as possible (also considering some of the reviewers' comments). I highlighted a couple points (see lines 312, 322, 336), but the authors should double-check again the whole manuscript.

Thank you for your remarks. We have clarified our wording throughout the manuscript.

Minor edits:

Lines 233-235: I would suggest changing “goes beyond those that measure biomass or assess single taxa to reveal drivers of insect communities” with something that highlights the advantages of the present study instead of pointing against other works (that “those”).

Something on the lines of:

“Our approach provides novel data on species richness across independent gradients of land-use intensity and climate. Furthermore, by combining malaise traps and DNA-metabarcoding, our work is not limited to single factors such as biomass measurements or assessment of single taxa to reveal drivers of insect communities.”

Thank you for your comment. We changed the sentence accordingly (p. 12, l. 238).

Lines 240-241: I would remove “Hallmann et al.” and just use “elsewhere”.

Changed accordingly

Lines 242-247: As I suggested above, I would recommend not using this kind of comparison, which may be perceived as stating other studies are “wrong”. The authors should focus on the perks of their study, and how that changes from others. The authors should rephrase this sentence stating what they found (or did not found) and THEN compare that with the results showed by other studies.

For example: “Our study recorded a temporal decline in biomass between the agricultural and semi-natural landscape types of only XXX. This appears to be in contrast with the results recorded in a similar analysis (Hallmann et al.) which showed a temporal decline in biomass of YYY in small, protected areas surrounded by an agricultural landscape. On the other hand, the variation in total species richness, matched the magnitude of the temporal decline (~35%) determined over a decade in grasslands and forests by Seibold et al.¹³”

This kind of rephrasing shift the focus from other studies to your own (which is more important!).

Thank you for this very helpful comment. This paragraph was also brought up by Reviewer 1. We changed the text according to your suggestions, see comment above.

Line 306: remove “very”.

Changed accordingly

Line 312: The authors should remember to specify which “biomass” or “richness” measurement they used and state it every time they want to discuss it. If you are referring to biomass variation across different areas, then you should specify it. If you have measured BIN richness, then you should specify it.

For example, “Across all habitats, biomass was best explained by..” with “Across all habitats, biomass variation(?) was best explained by..”

Changed accordingly

Same at line 322, 336 for biomass and richness. Should read “biomass variation” and “BIN richness”. Please, check the entire manuscript.

Changed accordingly

Line 347: “Country-wide strategy”

Corrected

Line 365: Reword to: “Nevertheless, additional studies should focus on biomes other than the cultivated landscapes of the temperate zone, such as cold boreal, dry Mediterranean, or hot tropical areas. Here, the different characteristics of the biome may result in land-use intensification being of less importance than climate change.”

We rephrased the paragraph accordingly (p. 18, l. 371).